# Cholesterol taste avoidance in *Drosophila melanogaster*

Roshani Nhuchhen Pradhan[1], Craig Montell[2], Youngseok Lee[1]*

[1]Department of Bio and Fermentation Convergence Technology, Kookmin University, Seoul, Republic of Korea; [2]Department of Molecular, Cellular, and Developmental Biology, and the Neuroscience Research Institute, University of California, Santa Barbara, Santa Barbara, United States

## eLife Assessment

This **useful** study provides **convincing** evidence that *Drosophila* can taste cholesterol through a subset of bitter-sensing gustatory receptor neurons, and that flies avoid high-cholesterol food. However, the same receptors have been previously found to be involved in the detection of multiple seemingly unrelated chemicals, and the reported expression patterns of these receptors contradict past reports. These caveats are not mentioned in the paper, raising critical concerns about the study's conclusions.

*For correspondence: ylee@kookmin.ac.kr

Competing interest: The authors declare that no competing interests exist.

**Abstract** The question as to whether animals taste cholesterol taste is not resolved. This study investigates whether the fruit fly, *Drosophila melanogaster*, is capable of detecting cholesterol through their gustatory system. We found that flies are indifferent to low levels of cholesterol and avoid higher levels. The avoidance is mediated by gustatory receptor neurons (GRNs), demonstrating that flies can taste cholesterol. The cholesterol-responsive GRNs comprise a subset that also responds to bitter substances. Cholesterol detection depends on five ionotropic receptor (IR) family members, and disrupting any of these genes impairs the flies' ability to avoid cholesterol. Ectopic expressions of these IRs in GRNs reveals two classes of cholesterol receptors, each with three shared IRs and one unique subunit. Additionally, expressing cholesterol receptors in sugar-responsive GRNs confers attraction to cholesterol. This study reveals that flies can taste cholesterol, and that the detection depends on IRs in GRNs.

## Introduction

Many types of tastants are beneficial at low concentrations, and harmful at high levels. Examples include minerals such as $Na^+$ and $Ca^{2+}$ (*Zhang et al., 2013a*, *Lee et al., 2018*), and fatty acids such as hexanoic acid (*Ahn et al., 2017*; *Pradhan et al., 2023*). Organic molecules such as cholesterol play essential roles in cellular membrane integrity, signaling functions, and steroid hormone synthesis. Cholesterol is a vital molecule, which supports numerous biological processes in animals, including reproduction, nutrient transport, and cellular activation (*Igarashi et al., 2018*). However, excessive cholesterol consumption can lead to a host of poor health consequences, including cardiovascular disease, and type 2 diabetes (*Soliman, 2018*). Due to the bivalent impact of cholesterol on human health, it stands to reason that there may be mechanisms that exist to promote or repress the taste of cholesterol. However, it is not clear whether cholesterol is sensed by the mammalian taste system. Mice and humans express several dozen taste receptors, most of which function in bitter taste (referred to as either T2Rs or TAS2Rs). The activities of two of these bitter receptors, T2R4 and T2R14 have

been shown to be modulated by cholesterol. However, it is unclear if they contribute to the taste of cholesterol (*Pydi et al., 2016*; *Shaik et al., 2019*; *Kim et al., 2024*).

It is plausible that insects such as *Drosophila melanogaster* might display a gustatory attraction to low levels of cholesterol since unlike vertebrates, which can synthesize cholesterol internally, fruit flies must obtain sterols through their diet (*Clark and Block, 1959*; *Niwa and Niwa, 2011*; *Shaheen, 2020*). Insects acquire cholesterol primarily from plant-derived phytosterols or pre-existing cholesterol (*Jing and Behmer, 2020*). For instance, *Manduca sexta* and *Bombyx mori* convert plant sterols to cholesterol through dealkylation in their gut, which is essential for producing hormones such as ecdysone (*Igarashi et al., 2018*). *Drosophila* acquires cholesterol directly from dietary sources such as phytosterols (e.g. sitosterol, stigmasterol) and fungal sterols (e.g. ergosterol) found in yeast (*Niwa and Niwa, 2011*). Given that consumption of high cholesterol is harmful (*Soliman, 2018*; *Schade et al., 2020*), fruit flies might display a gustatory aversion to high levels, while finding low levels attractive. Such a bivalent response would be similar to the flies' taste attraction to low concentrations of $Na^+$ and their repulsion to high $Na^+$ (*Zhang et al., 2013a*, *Jaeger et al., 2018*; *Taruno and Gordon, 2023*; *Sang et al., 2024*). $Ca^{2+}$ is also required at low levels and is deleterious at high concentrations. We have previously shown that fruit flies are indifferent to modest levels of $Ca^{2+}$ and avoid high $Ca^{2+}$ (*Lee et al., 2018*). Thus, if flies are endowed with the capacity to taste cholesterol, it is open question as to whether they would have a bivalent gustatory response depending on concentration, similar to $Na^+$ (*Zhang et al., 2013a*, *Jaeger et al., 2018*; *Taruno and Gordon, 2023*; *Sang et al., 2024*), or be indifferent to low cholesterol and reject high cholesterol, similar to the flies' differential reaction to $Ca^{2+}$ depending on concentration (*Lee et al., 2018*).

In *Drosophila*, gustatory organs are distributed on multiple body parts, including the labellum at the end of the proboscis, which represents the largest taste organ. The end of the proboscis is endowed with two labella, each of which is decorated with 31 external bristles. These sensilla house either two or four GRNs, which respond to external chemical stimulation and modulate behavioral responses (*Dahanukar et al., 2001*; *Larsson et al., 2004*; *Suh et al., 2004*; *Benton et al., 2009*; *Cameron et al., 2010*; *Chen et al., 2010*; *Kim et al., 2010*; *Kwon et al., 2010*; *Rimal and Lee, 2018*). This is accomplished through expression of a diverse repertoire of receptor classes, including gustatory receptors (GRs), IRs, pickpocket (PPK) ion channels, and transient receptor potential (TRP) channels.

In this work, we reveal that flies taste cholesterol. Reminiscent of their reaction to $Ca^{2+}$ (*Lee et al., 2018*), they are indifferent to low cholesterol and reject high cholesterol. Using a combination of behavioral and electrophysiological assays, we demonstrate that a subset of the same class of GRNs that responds to bitter chemicals is also required in adults to avoid the taste of higher cholesterol levels. In addition, we found that multiple members of the IR family are involved in cholesterol taste perception, and that there are two overlapping sets of IRs that are sufficient to confer cholesterol sensitivity to GRNs that normally do not respond to cholesterol. This work establishes that flies can taste cholesterol, and defines the underlying cellular and molecular mechanisms involved in rejection of high cholesterol.

## Results
### Flies taste cholesterol through a subset of bitter GRNs

To address whether fruit flies can taste cholesterol, we investigated whether cholesterol triggers action potentials in GRNs associated with taste bristles in the labella. The 31 sensilla present in each labellum are categorized into long (L), intermediate (I), and short (S) subtypes (*Figure 1A*; *Hiroi et al., 2002*). To examine cholesterol-induced action potentials in response to a range of cholesterol concentrations, we focused on S7, I8, and L6 sensilla and performed tip recordings. We detected action potentials in S7 once the cholesterol concentration reached $10^{-3}$ %, whereas the I8 and L6 were nearly unresponsive even at 0.1% (*Figure 1B and C*). The methyl-β-cyclodextrin (MβCD) used to dissolve cholesterol did not evoke spikes (*Figure 1—figure supplement 1A and B*). We then analyzed all 31 sensilla using 0.1% cholesterol. We found that the S-type sensilla, especially the S6 and S7 sensilla, were most responsive, while very few spikes were induced from the I-type or L-type sensilla (*Figure 1D*).

To determine which GRN type contributes to cholesterol-induced action potentials, we selectively inactivated different classes of GRNs by expressing a transgene encoding the inwardly rectifying potassium channel, Kir2.1 (*Baines et al., 2001*). The bristles on the labellum harbor GRNs

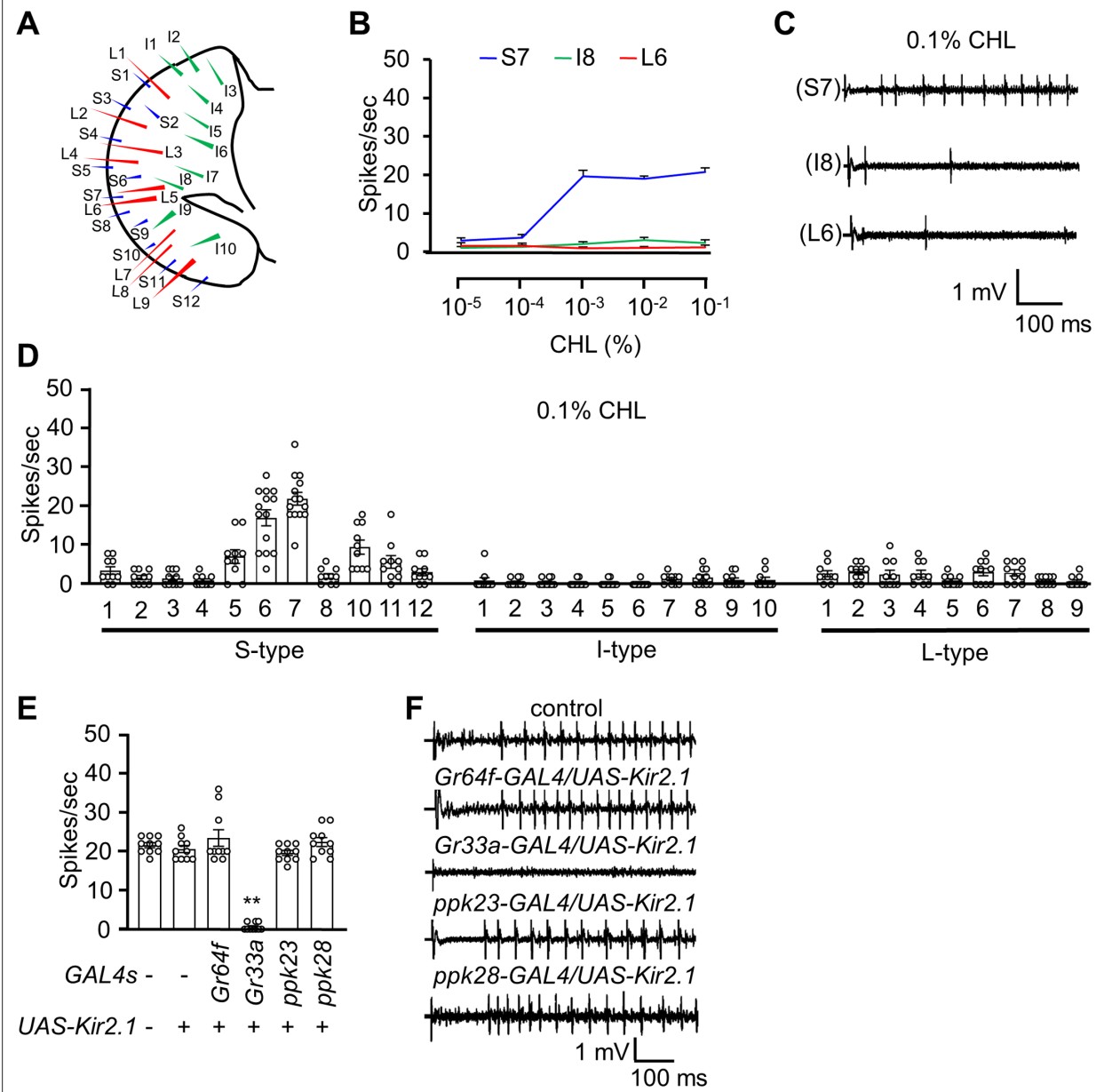

**Figure 1.** The neuronal response of the adult flies to cholesterol. (**A**) Schematic diagram of the fly labellum. (**B**) Average frequencies of action potential generated from S7, I8, and L6 sensilla upon application of different concentrations of cholesterol (CHL; n=10–12). (**C**) Representative sample traces of S7, I8, and L6 from (**B**). (**D**) Electrophysiological responses of control flies produced from all labellum sensilla in response to 0.1% cholesterol (n=10–12). (**E**) Electrophysiological analysis of S7 sensilla in response to 0.1% cholesterol using flies in which different GRNs were inactivated by the inwardly rectifying potassium channel *Kir2.1* (n=10–12). (**F**) Representative sample traces of the S7 sensilla from (**E**). All error bars represent SEMs. Single-factor ANOVA was combined with Scheffe's post hoc analysis to compare multiple datasets. Asterisks indicate statistical significance compared to the control group (**p<0.01).

The online version of this article includes the following figure supplement(s) for figure 1:

**Figure supplement 1.** Electrophysiological responses using different doses of methyl-β-cyclodextrin (MβCD).

that fall into four main classes, each of which expresses a gene driver unique to that class (**Montell, 2021**). These include A GRNs, which respond to sugars, low Na⁺ and other attractive compounds (*Gr64f-GAL4*), B GRNs, which are stimulated by bitter compounds, high Na⁺ and other aversive chemicals (*Gr33a-GAL4*), C GRNs, which are activated by $H_2O$ and hypo-osmolarity (*ppk28-GAL4*), and D GRNs, which respond to $Ca^{2+}$ and high concentrations of other cations (*ppk23-GAL4*) (**Thorne et al., 2004**; **Dahanukar et al., 2007**; **Moon et al., 2009**; **Cameron et al., 2010**; **Lee et al., 2018**). We found

that silencing B GRNs reduced neuronal responses to cholesterol, whereas inhibition of other GRN types exhibited normal neuronal firing (*Figure 1E and F*). These data demonstrate that cholesterol is sensed by B GRNs in the labellar sensilla.

## A cluster of IRs is required to sense cholesterol in adult *Drosophila*

To pinpoint the molecular sensors for detecting cholesterol, we first investigated requirements for the largest family of taste receptors–the GRs. Six GRs are broadly expressed in bitter GRNs and three of them serve as co-receptors (*Montell, 2021*; *Shrestha and Lee, 2023*), including GR32a (*Miyamoto and Amrein, 2008*), GR33a (*Lee et al., 2009*; *Moon et al., 2009*), GR39a.a (*Dweck and Carlson, 2020*), GR66a (*Moon et al., 2006*), GR89a (*Shrestha and Lee, 2021a*), and GR93a (*Lee et al., 2009*). We performed tip recordings, demonstrating that mutations disrupting any of these co-receptors had no impact on cholesterol-induced action potentials (*Figure 2—figure supplement 1A*). *Drosophila* encodes 13 TRP channels, several of which function in taste (*Al-Anzi et al., 2006*; *Kang et al., 2010*; *Kim et al., 2010*; *Zhang et al., 2013b*; *Mandel et al., 2018*; *Leung et al., 2020*; *Montell, 2021*; *Gong et al., 2004*). We analyzed mutant lines disrupting most of these channels and found that the neuronal responses were normal (*Figure 2—figure supplement 1B*).

IRs comprise another large family of receptors that function in taste, as well as in other sensory processes (*Rytz et al., 2013*; *Rimal and Lee, 2018*). To address whether any IR is required for cholesterol taste, we screened the 32 available *Ir* mutants by performing tip recording on S7 sensilla using 0.1% cholesterol. Most mutants displayed normal responses (*Figure 2A*), including those with previously identified gustatory functions such as *Ir7a$^1$* (acetic acid sensor) (*Rimal et al., 2019*), *Ir7c$^{GAL4}$* and *Ir60b$^3$* (high Na$^+$) (*McDowell et al., 2022*; *Sang et al., 2024*), *Ir56b$^1$* (low Na$^+$) (*Dweck et al., 2022*), *Ir62a$^1$* (Ca$^{2+}$) (*Lee et al., 2018*), *Ir94f$^1$* (cantharidin) (*Pradhan et al., 2024*) as well as *Ir20a$^1$, Ir47a$^1$, Ir52a$^1$,* and *Ir92a$^1$* (alkali) (*Pandey et al., 2023*). In contrast, our survey revealed that five mutants (*Ir7g$^1$, Ir25a$^2$, Ir51b$^1$, Ir56d$^1$,* and *Ir76b$^1$*) exhibited strong defects in firing in response to cholesterol (*Figure 2A*). Two of the mutations, *Ir25a$^2$* and *Ir76b$^1$*, disrupt co-receptors that are necessary for sensing most attractive and aversive tastants (*Ganguly et al., 2017*; *Lee et al., 2018*; *Dhakal et al., 2021*; *Shrestha and Lee, 2021b, Stanley et al., 2021*; *Aryal et al., 2022a, Xiao et al., 2022*; *Li et al., 2023*; *Pandey et al., 2023*; *Pradhan et al., 2024*). In further support of the roles of these five IRs for detecting cholesterol, we observed similar phenotypes resulting from mutation of additional alleles (*Ir7g$^2$, Ir51b$^2$, Ir56d$^2$,* and *Ir76b$^2$*) (*Zhang et al., 2013a, Sánchez-Alcañiz et al., 2018*; *Dhakal et al., 2021*; *Pradhan et al., 2024*), or due to placing the mutation (*Ir25a$^2$*) in trans with a deficiency (Df) spanning the locus (*Figure 2B*). Furthermore, using the phytosterol stigmasterol to stimulate S6, S7, and S10 sensilla, we confirmed that the five mutants exhibited consistent phenotypes, underscoring the specificity of these IRs for sterol detection (*Figure 2—figure supplement 1C-E*).

To provide additional verification that the phenotypes exhibited by the *Ir7g$^1$, Ir25a$^2$, Ir51b$^1$, Ir56d$^1$,* and *Ir76b$^1$* mutants were attributed to the loss of *Ir7g, Ir25a, Ir51b, Ir56d,* and *Ir76b*, we conducted rescue experiments. To do so, we used *GAL4* lines specific to each gene (*Ir25a, Ir56d,* and *Ir76b*) or *Gr33a-GAL4* to drive the respective wild-type *UAS-cDNA*s in the corresponding mutant backgrounds. We found that the responses to cholesterol were fully restored in S7 sensilla stimulated with 0.1% cholesterol (*Figure 2D and F*).

To evaluate the dose-dependent defects exhibited by the mutants, we performed tip recordings to examine the neuronal responses of S7 sensilla to a spectrum of cholesterol percentages ($10^{-5}$ to $10^{-1}$). All five mutants exhibited significantly reduced neuronal firing in response to cholesterol percentages over a 100-fold range ($10^{-3}$ to $10^{-1}$; *Figure 2E*). However, at lower percentages of cholesterol ($10^{-5}$ and $10^{-4}$), all five IR mutants did not vary significantly from the control (*Figure 2E*).

## IRs required in B GRNs for cholesterol-induced neuronal firing

To test whether the IRs function in B GRNs, we used two approaches: RNA interference (RNAi) and gene rescue experiments. To knock down gene expression in B GRNs, we took advantage of the *Gr33a-GAL4* and found that targeting any of the five genes dramatically reduced action potentials in S7 sensilla in response to 0.1% cholesterol (*Figure 2C*). In contrast, when we used a D GRN driver (*ppk23-GAL4*) in combination with the same *UAS-RNAi* lines, there was no decrease in neuronal firing. To perform gene rescue experiments, we used the *Gr33a-GAL4* to express each wild-type cDNA

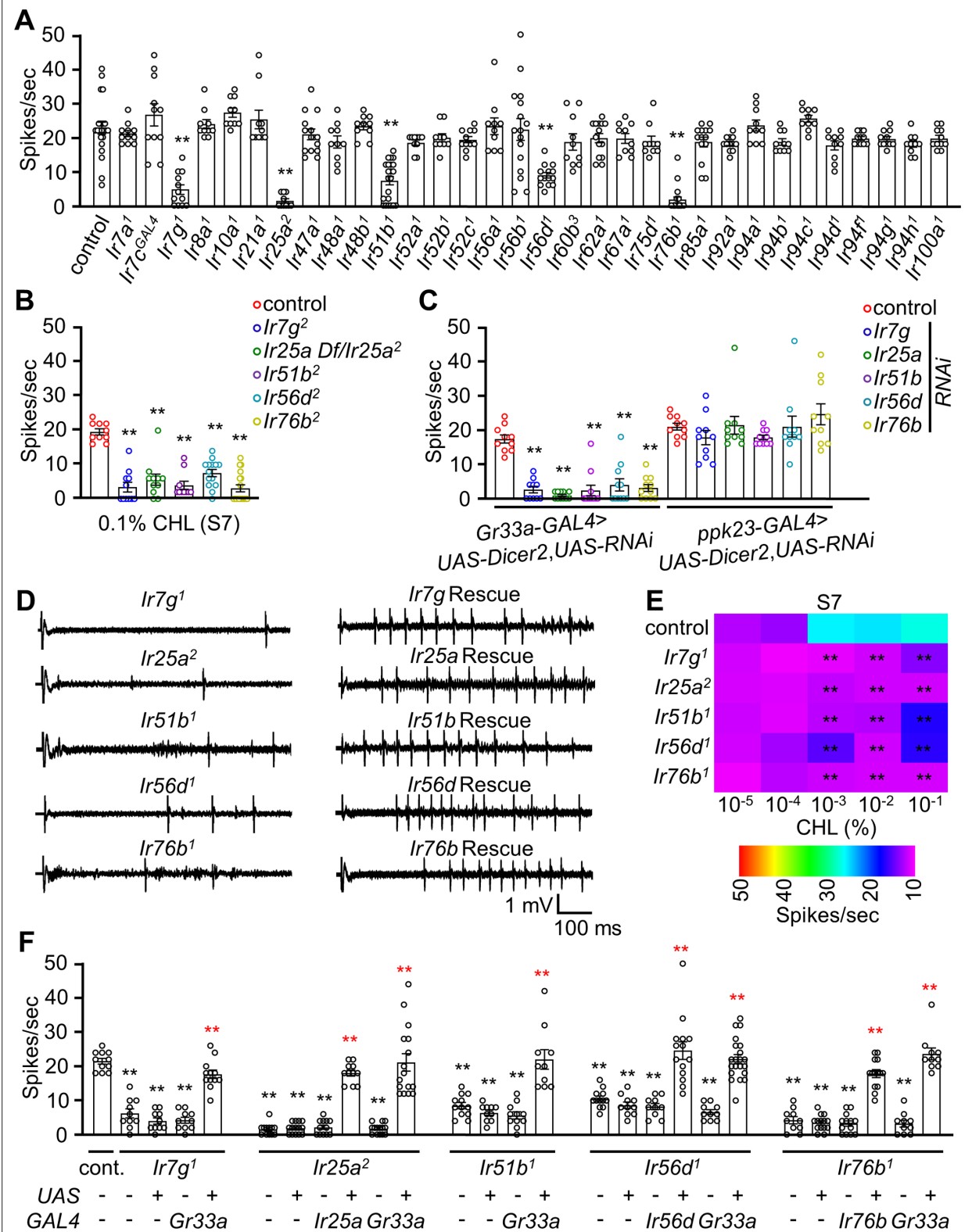

**Figure 2.** Ionotropic receptors (IRs) are responsible for sensing cholesterol. (**A**) Tip recordings using 0.1% cholesterol to analyze the responses of S7 sensilla from control flies and from 32 *Ir* mutants (n=10–16). (**B**) Tip recordings using 0.1% cholesterol to analyze responses of S7 sensilla from *Ir7g²*, *Ir25a Df/Ir25a²*, *Ir51b²*, *Ir56d²*, and *Ir76b²* (n=10–16). (**C**) Tip recordings using 0.1% cholesterol to analyze responses of S7 sensilla after RNAi knockdown of the following genes using either the *Gr33a-GAL4* or *ppk23-GAL4*: *Ir7g*, *Ir25a*, *Ir51b*, *Ir56d*, and *Ir76b*. (**D**) Representative sample traces of (**F**) for control,

*Figure 2 continued on next page*

*Figure 2 continued*

mutants, and rescue lines using the *GAL4/UAS* system. (**E**) Heatmap representing the dose responses (spikes/sec) elicited by S7 sensilla from the control and the indicated mutants (*Ir7g¹*, *Ir25a²*, *Ir51b¹*, *Ir56d¹*, and *Ir76b¹*) (n=10–16). (**F**) Tip recordings performed on S7 sensilla (0.1% cholesterol) from control, *Ir7g¹*, *Ir25a²*, *Ir51b¹*, *Ir56d¹*, *Ir76b¹*, and from flies expressing the indicated cognate transgenes under control of either their own *GAL4* or the *Gr33a-GAL4* (n=10–14). All error bars represent SEMs. Single-factor ANOVA was combined with Scheffe's post hoc analysis to compare multiple datasets. Black asterisks indicate statistical significance compared to the control group. The red asterisks indicate statistical significance between the control and the rescued flies (**p<0.01).

The online version of this article includes the following figure supplement(s) for figure 2:

**Figure supplement 1.** Electrophysiological analyses of S7 sensilla from mutants disrupting different bitter GRs and TRP channels in the presence of $10^{-1}$% CHL, and a subset of bitter GRNs express *Ir56d*.

transgene in the corresponding mutant background and performed tip recordings. In all cases, we rescued the mutant phenotypes (*Figure 2D and F*). Thus, we conclude that the IRs function in B GRNs.

Given that the five IRs are required in B GRNs, it stands to reason that they are expressed in these neurons. Indeed, *Ir7g*, *Ir25a*, *Ir51b*, and *Ir76b* have been shown previously to be expressed in B GRNs (*Lee et al., 2018*; *Dhakal et al., 2021*; *Pradhan et al., 2024*). However, *Ir56d*, which has a role in sweet-sensing A GRNs, has not. To explore this possibility, we performed double-labeling experiments. We expressed *UAS-dsRed* under the control of the *Ir56d-GAL4 and* did so in files that included a B GRN reporter (*Gr66a*-I-*GFP*). Each of the two bilaterally symmetrical labella contain 11 S-type sensilla, 11 I-type sensilla, and 9 L-type sensilla. We found that 10.7±1.4 cells co-expressed both the dsRed and GFP markers (*Figure 2—figure supplement 1F*). By tracing dendrites from individual GFP-expressing cells, we identified the specific sensilla innervated by each marker. Most S2, S3, S4, S6, and S7 sensilla that expressed the *Ir56d* reporter were co-labeled with the B GRN reporter. Thus, the B GRNs in the two sensilla that elicited the highest frequency of cholesterol-induced action potentials (S6 and S7) were labeled by the *Ir56d* reporter.

## IRs required for avoiding the taste of cholesterol

The requirement for the five IRs for cholesterol-induced action potentials in B GRNs suggests that cholesterol is an aversive taste. To explore this question, we used the well-established binary choice assay in which we allowed flies to choose between 2 mM sucrose alone or 2 mM sucrose mixed with various percentages of cholesterol. We mixed the two food alternatives with either blue or red food dye so we could inspect the flies' abdomens to assess which option they consumed (*Aryal et al., 2022b*). At the lowest percentage tested ($10^{-5}$%), flies showed only a slight aversion to cholesterol-containing food (*Figure 3A*). As cholesterol concentration increased, they showed a dose-dependent aversion, with a very strong aversion at 0.1% (PI = −0.72 ± 0.03; *Figure 3A*). Both male and female flies showed comparable avoidance responses to 0.1% cholesterol, indicating the behavior is not sex-specific (*Figure 3B*). The aversion was not due to the MβCD used to dissolve the cholesterol since the flies were indifferent to sucrose alone versus sucrose plus MβCD (*Figure 3—figure supplement 1A*). Moreover, the flies showed similar levels of aversion to sucrose plus cholesterol versus either sucrose alone (*Figure 3A*) or sucrose plus MβCD (*Figure 3—figure supplement 1B*). The dyes used in the study also did not alter the behavioral response (*Figure 3—figure supplement 1C*).

To determine the impact of inhibiting B GRNs on gustatory behavior, we used the *Gr33a-GAL4* to drive an expression of *UAS-Kir2.1*. As a control, we also inactivated other classes of GRNs and found that expression of *kir2.1* in A GRNs (*Gr64f-GAL4*), C GRNs (*ppk23-GAL4*), and D GRNs (*ppk28-GAL4*) had no impact on cholesterol avoidance (*Figure 3C*). Surprisingly, inhibiting B GRNs (*Gr33a-GAL4*) not only eliminated cholesterol avoidance, it caused the flies to exhibit a preference for cholesterol-containing food, thereby unmasking some unknown attractive mechanism.

We also set out to assess the requirements for the five IRs for cholesterol taste. Therefore, we performed two-way choice assays. All mutants showed defects in avoiding the sucrose-containing 0.1% cholesterol over sucrose alone (*Figure 3D and E*). We then tested the behavior of the mutants across a range of cholesterol percentages ($10^{-5}$ to $10^{-1}$%). We found that the *Ir25a²*, *Ir51b¹*, *Ir56d¹*, and *Ir76b¹* mutants exhibited reduced aversion across all cholesterol concentrations tested (*Figure 3F*). However, *Ir7g¹* showed a deficit in behavioral avoidance only at higher cholesterol percentages ($10^{-2}$ and $10^{-1}$; *Figure 3F*). To test for rescue, we expressed the wild-type cDNA of each IR using its

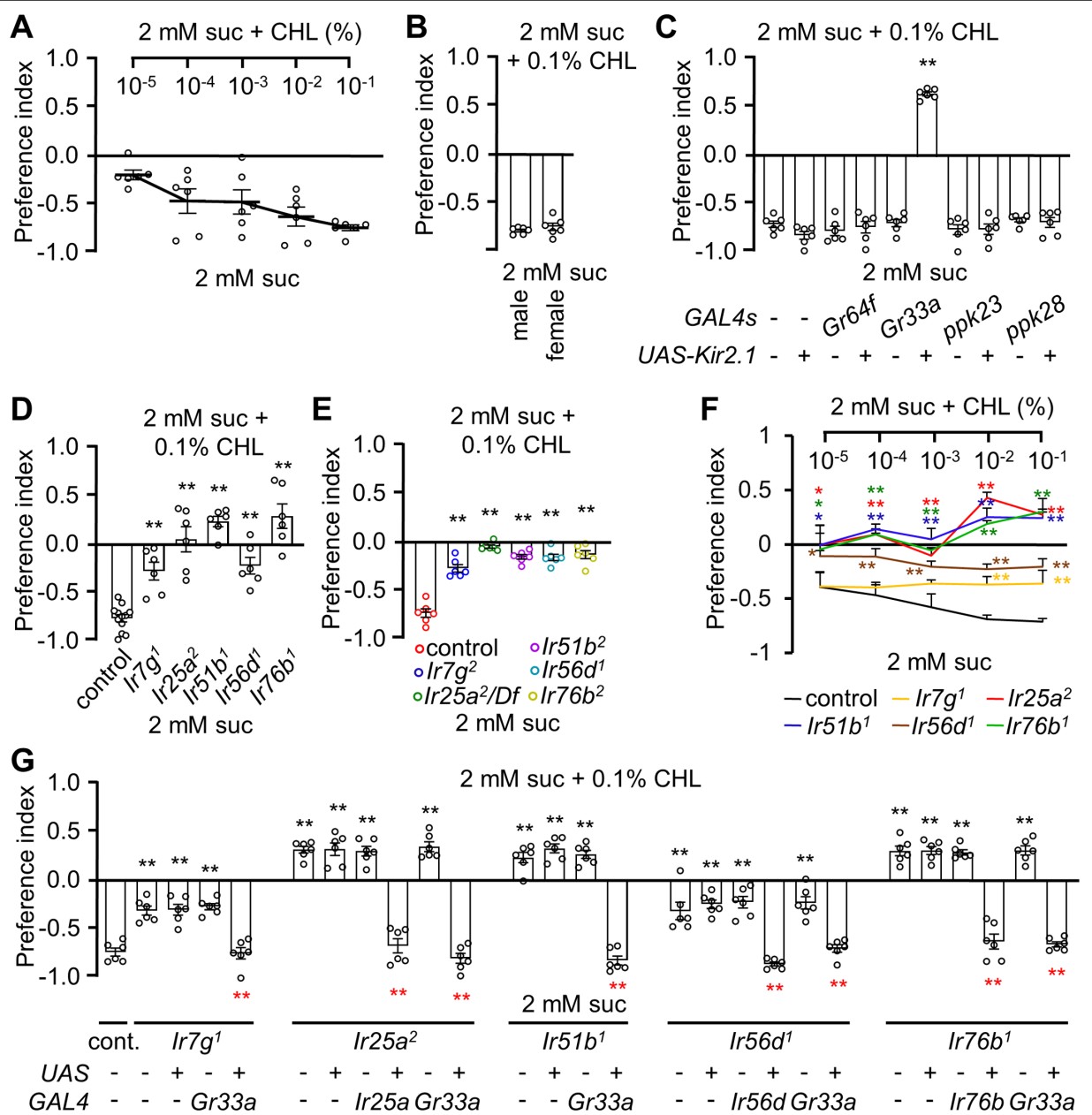

**Figure 3.** *Ir7g*, *Ir25a*, *Ir51b*, *Ir56d*, and *Ir76b* are required for the perception of cholesterol. (**A**) Binary food choice analysis of *w1118* adult flies toward different doses of cholesterol. Sucrose (2 mM) was included on both sides (n=6). (**B**) Binary food choice analyses to test for sex-specific difference in the feeding responses toward 0.1% cholesterol (n=6). (**C**) Binary food choice assays to determine the effects of inactivating different GRN types on the responses to 0.1% cholesterol. +/-indicates the presence or absence of the transgene, respectively (n=6). (**D**) Binary food choice assays to test the reponses of *Ir7g1*, *Ir25a2*, *Ir51b1*, *Ir56d1*, and *Ir76b1* flies to 0.1% cholesterol (n=6). (**E**) Binary food choice assays to analyze the responses of *Ir7g2*, *Ir25a Df*, *Ir51b2*, *Ir56d2*, and *Ir76b2* flies to 0.1% cholesterol (n=6). (**F**) Dose responses of control, *Ir7g1*, *Ir25a2*, *Ir51b1*, *Ir56d1*, and *Ir76b1* flies to different concentrations of cholesterol (10−5%, 10−4%, 10−3%, 10−2%, and 10−1%) via binary food choice assays (n=6). (**G**) Rescue of *Ir7g1*, *Ir25a2*, *Ir51b1*, *Ir56d1*, and *Ir76b1* defects by expressing the wild-type cDNAs under the control of the *GAL4* drivers specific to each gene (*Ir25a*, *Ir56d*, and *Ir76b*) or *Gr33a-GAL4* (n=6). All error bars represent SEMs. Single-factor ANOVA was combined with Scheffe's post hoc analysis to compare multiple datasets. Black asterisks indicate statistical significance compared to the control group. The red asterisks indicate statistical significance between the control and the rescued flies (**p<0.01).

The online version of this article includes the following figure supplement(s) for figure 3:

**Figure supplement 1.** Binary food choice assays with CHL and methyl-β-cyclodextrin (MβCD).

respective *GAL4* (*Ir25a*, *Ir56d*, and *Ir76b*) or *Gr33a-GAL4*. The avoidance deficiencies in the five IR mutants were fully restored (*Figure 3G*).

To determine whether the IRs function in B GRNs, we performed rescue experiments and RNAi. In all cases, the mutant phenotypes were rescued using the B GRN driver (*Gr33a-GAL4*) in combination with the corresponding *UAS-cDNA* (*Figure 3G*). To perform RNAi, we also took advantage of the *Gr33a-GAL4*. We found that knockdown of each *Ir* in B GRNs eliminated cholesterol avoidance (*Figure 3—figure supplement 1D*). However, RNAi knockdown in D GRNs (*ppk23-GAL4*) had no impact on the repulsion to cholesterol (*Figure 3—figure supplement 1E*). Thus, we conclude that the *Ir*s function in B GRNs.

We also conducted binary food choice assays to address whether olfaction contributed to the avoidance of cholesterol. We found that the *orco* null mutant (*orco¹*), which disrupts the olfactory co-receptor (ORCO) broadly required for olfaction, exhibited cholesterol repulsion similar to control flies (*Figure 3—figure supplement 1F*). Consistent with these findings, surgically ablating the antennae and maxillary palps, which are the main olfactory organs, did not diminish cholesterol avoidance (*Figure 3—figure supplement 1G*).

## Ectopic co-expression of two sets of IRs confers responses to cholesterol

To explore whether IR7g, IR25a, IR51b, IR56d, and IR76b are sufficient to confer cholesterol sensitivity to GRNs that are normally unresponsive to cholesterol, we conducted ectopic expression experiments. We expressed the five *Ir*s in all B GRNs (*Gr33a-GAL4*; *Figure 4A*) or in all A GRNs (*Gr5a-GAL4*; *Figure 4C*) and then characterized cholesterol-induced action potentials by focusing on cholesterol-insensitive I-type sensilla. Introducing all five *Ir*s into cholesterol-insensitive, I9 sensilla elicited strong responses (*Figure 4B*). Misexpression of just *Ir7g*, *Ir51b*, and *Ir56d* also replicated cholesterol-induced responses (*Figure 4B*), presumably because *Ir25a* and *Ir76b* are endogenously expressed in GRNs in these sensilla (*Lee et al., 2018*). Expression of any one of these *Ir*s (*Ir7g*, *Ir51b*, and *Ir56d*) or combining *Ir7g* and *Ir51b* was insufficient to induce cholesterol sensitivity (*Figure 4B*). Of significance, we found that combining *Ir56d* with either *Ir51b* or *Ir7g* conferred cholesterol sensitivity to I9 sensilla (*Figure 4B*).

L-type sensilla are missing B GRNs and are unresponsive to cholesterol. Therefore, we misexpressed all five *Ir*s in A GRNs using the *Gr5a–GAL4* (*Figure 4C*) and characterized action potentials in L6 sensilla. Ectopic expression of the *Ir*s in the A GRNs also bestowed responsiveness to cholesterol (*Figure 4D*). Consistent with the results in B GRNs, co-expression of *Ir51b* and *Ir56d*, or *Ir7g* and *Ir56d*, was sufficient to confer cholesterol sensitivity (*Figure 4D*). This indicates that either of two groups of four *Ir*s (*Ir7g*, *Ir25a*, *Ir56d*, *Ir76b* or *Ir25a*, *Ir51b*, *Ir56d*, *Ir76b*) is sufficient to comprise a functional cholesterol receptor.

## Inducing attraction to cholesterol

Activation of A GRNs by sugars and several other attractive chemicals promotes feeding. Given that ectopic expression of *Ir7g* and *Ir56d*, or *Ir51b* and *Ir56d* alone was sufficient to induce a response to cholesterol in A GRNs, we investigated whether this would elicit attraction toward cholesterol. Control flies exhibited a preference for 2 mM sucrose alone over 2 mM sucrose laced with cholesterol (*Figure 4E*). *Ir56d¹* and *Ir7g¹* mutant flies showed a slight avoidance of cholesterol-laced food. Flies carrying the *Ir56d¹* or the *Ir7g¹* mutation and expressing both *UAS-Ir7g* and *UAS-Ir51b* in A GRNs exhibited similar behavior as *Ir56d¹* or *Ir7g¹* flies. However, when we introduced *Ir7g* and *Ir56d*, or *Ir51b* and *Ir56d* in the mutants, the flies exhibited attraction to the cholesterol-laced food (*Figure 4E*).

## Discussion

The impact of cholesterol on an animal's health depends on the concentration of cholesterol that is consumed (*Huang et al., 2024*). While low levels are crucial, animals must avoid consuming excessive cholesterol (*Beynen, 1988*; *Soliman, 2018*; *Zhang et al., 2019*). This differential effect of cholesterol is reminiscent of the impact of $Na^+$ and $Ca^{2+}$ on health, depending on their concentration (*Zhang et al., 2013a*, *Lee et al., 2018*). Flies have a bivalent reaction to $Na^+$ depending on concentration but only avoid high $Ca^{2+}$ and are indifferent to low $Ca^{2+}$. Therefore, it was an open question as to whether

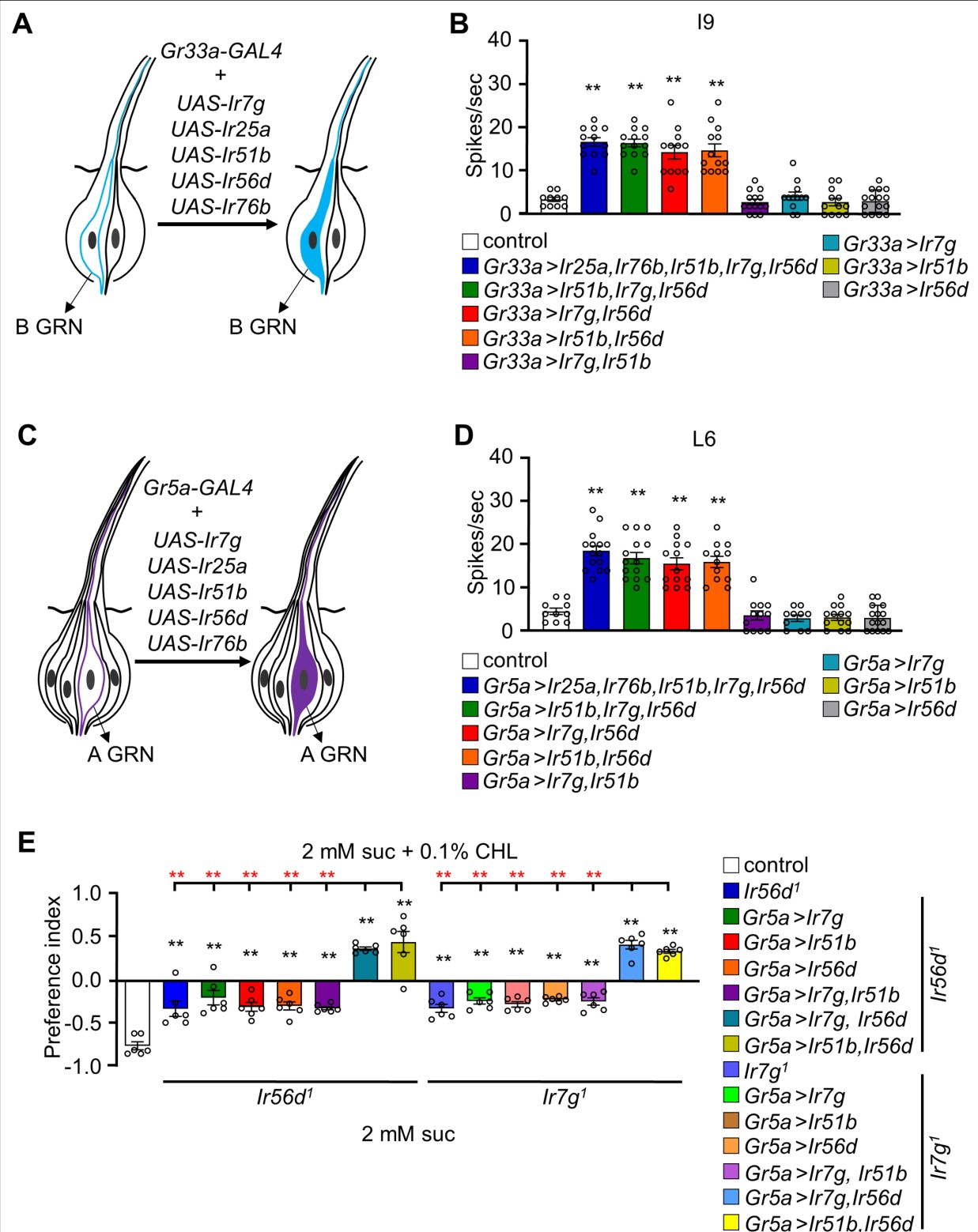

**Figure 4.** Testing whether ectopic expression of *Ir7g, Ir25a, Ir51b, Ir56d,* and *Ir76b* in L- and I-type sensilla confers cholesterol responsiveness. (**A**) Schematic representation of ectopic expression of *Ir*s in B GRNs under control of the *Gr33a-GAL4*. (**B**) Tip recordings conducted from I9 sensilla with 0.1% cholesterol using flies overexpressing *UAS-Ir7g, UAS-Ir25a, UAS-Ir51b, UAS-Ir56d,* and *UAS-Ir76b* in B GRNs under control of the *Gr33a-GAL4* (n=10–16). (**C**) Schematic presentation of misexpression of *Ir*s in A GRNs under control of the *Gr5a-GAL4*. (**D**) Tip recordings from L6 sensilla of the indicated flies expressing the indicated *Ir*s under control of the *Gr5a-GAL4* (n=10–16). (**E**) Binary food choice assays testing for attraction or aversion

*Figure 4 continued on next page*

*Figure 4 continued*

to 0.1% cholesterol in flies misexpressing *Ir7g*, *Ir51b*, and *Ir56d* in A GRNs (*Gr5a-GAL4*). The *Ir*s were ectopically expressed in either an *Ir56d*[1] or *Ir7g*[1] mutant background (n=6). The red asterisks indicate the comparison of the combination of two *UAS* lines (*Ir7g, Ir56d* and *Ir51b, Ir56d*) driven by *Gr5a-GAL4* with all the single *UAS* line including the combination of *Ir7g* and *Ir51b*. All error bars represent SEMs. Single-factor ANOVA was combined with Scheffe's post hoc analysis to compare multiple datasets. Black asterisks indicate statistical significance compared with the control (**$p<0.01$).

flies have the capacity to taste cholesterol, and if so, whether they are endowed with the capacity to respond differentially to low and high concentrations or only avoid high cholesterol.

Several biochemical and biophysical studies focusing on the mammalian taste receptors T2R4 and T2R14 demonstrate that these receptors can bind to and be activated or modulated by cholesterol (*Pydi et al., 2016*; *Shaik et al., 2019*; *Kim et al., 2024*). While T2R4 and T2R14 are expressed in the taste system, they are expressed at high levels extraorally, such as in airway epithelial cells, pulmonary artery smooth muscle cells, and breast epithelial cells (*Hariri et al., 2017*; *Jaggupilli et al., 2017*; *Singh et al., 2020*). Therefore, it has been thought that these receptors may function in interoception, enabling the body to sense and respond to internal levels of cholesterol. Currently, evidence that these or other receptors function in cholesterol taste is lacking in mammals or any other animal.

Our research highlights the discovery that flies reject higher levels of cholesterol, but do not show attraction to low cholesterol even though flies cannot synthesize cholesterol and, therefore, must meet their needs for cholesterol from their diet. Nevertheless, the repulsion to a required substance is reminiscent of the fly's response to $Ca^{2+}$, which is aversive even though $Ca^{2+}$ is required for life (*Lee et al., 2018*). Moreover, we discovered that the repulsion to high cholesterol is mediated through the taste system since cholesterol stimulates action potentials in a subset of GRNs. The class of GRN that is activated by cholesterol is the B class, which also responds to bitter chemicals and other aversive tastants. However, cholesterol stimulates only a subset of bitter-responsive GRNs.

An unexpected observation is that inhibition of B GRNs with Kir2.1 not only eliminates cholesterol repulsion but causes cholesterol to become highly attractive. This unmasking of an attractive mechanism for cholesterol when B GRNs are inhibited raises questions about the underlying neural circuitry and molecular mechanisms. Notably, this effect is unlikely to be due to cholesterol-induced activation of GRNs that promote feeding, since cholesterol does not activate any GRN in L-type sensilla, which are devoid of B GRNs, but include three types of attractive GRNs: A GRNs, C GRNs, and other class of GRNs (E) (*Montell, 2021*). Thus, we suggest that the attraction to cholesterol, which is unmasked by inhibition of B GRNs, occurs through a mechanism postsynaptic to the GRNs. Understanding the attractive mechanisms could provide valuable insights into how *Drosophila* regulates cholesterol intake based on internal nutritional states. This is particularly relevant given that *Drosophila*, like other insects, are cholesterol auxotrophs and must obtain sterols from their diet (*Carvalho et al., 2010*) Furthermore, elucidating the neural and molecular basis of cholesterol attraction and its potential modulation by internal metabolic states in *Drosophila* has the potential to reveal evolutionarily conserved mechanisms.

A key issue concerns the molecular identity of the *Drosophila* cholesterol taste receptor. We addressed this question using both electrophysiological and behavioral approaches to assess the impact of mutating genes encoding receptors belonging to the major families of fly taste receptors. We found that five IRs are required for cholesterol taste. These include two broadly required co-receptors (IR25a and IR76b) and three other receptors (IR7g, IR51b, and IR56d). The contribution of five IRs to cholesterol taste was unanticipated since IRs are thought to be tetramers (*Wicher and Miazzi, 2021*). Therefore, we conducted a series of ectopic expression experiments to determine whether all five IRs were necessary to confer cholesterol sensitivity to GRNs that do not normally respond to cholesterol. We found that either of two combinations of four IRs was sufficient to endow cholesterol responsiveness to GRNs. Three of the IRs were common to both receptors (IR25a, IR56d, IR76b). However, addition of either IR7g or IR51b as the fourth IR was necessary to generate a cholesterol receptor. It remains to be determined as to why there are two cholesterol receptors, and why both are required for cholesterol taste.

The findings reported here raise the question as to whether mammals such as mice and humans perceive cholesterol through the sense of taste. It is notable that in flies cholesterol taste depends on

B GRNs, which also sense bitter compounds, and that in mammals T2Rs are activated by cholesterol, since this family of receptors also respond to bitter compounds. Therefore, it is intriguing to speculate that cholesterol taste may be aversive in humans.

## Materials and methods

### Key resources table

| Reagent type (species) or resource | Designation | Source or reference | Identifiers | Additional information |
|---|---|---|---|---|
| Genetic reagent (*Drosophila melanogaster*) | $Ir7a^1$ | **Rimal et al., 2019** | | Provided by Dr. Y. Lee |
| Genetic reagent (*Drosophila melanogaster*) | $Ir7g^1$: $y^1$ w* Mi{$y^{+mDint2}$=MIC}Ir7g$^{MI06687}$ | Bloomington *Drosophila* Stock Center | BDSC:42420 | |
| Genetic reagent (*Drosophila melanogaster*) | $Ir8a^1$:w[*]TI{w[+m*]=TI}$Ir8a$(1);Bl(1)L(2)/CyO | Bloomington *Drosophila* Stock Center | BDSC:23842 | |
| Genetic reagent (*Drosophila melanogaster*) | $Ir10a^1$:$w^{1118}$ Mi{GFP$^{E.3xP3}$=ET1}$Ir10a^{MB03273}$ | Bloomington *Drosophila* Stock Center | BDSC:41744 | |
| Genetic reagent (*Drosophila melanogaster*) | $Ir21a^1$: $w^{1118}$; PBac{$w^{+mC}$=PB}Ir21a$^{c02720}$ | Bloomington *Drosophila* Stock Center | BDSC:10975 | Provided by Dr. C. Montell |
| Genetic reagent (*Drosophila melanogaster*) | $Ir25a^2$ | **Benton et al., 2009** | | Provided by Dr. L. Voshall |
| Genetic reagent (*Drosophila melanogaster*) | $Ir47a^1$ | **Rimal et al., 2019** | | Provided by Dr. Y. Lee |
| Genetic reagent (*Drosophila melanogaster*) | $Ir48a^1$: $w^{1118}$; Mi{GFP$^{E.3xP3}$=ET1}$Ir48a^{MB09217}$ | Bloomington *Drosophila* Stock Center | BDSC:26453 | |
| Genetic reagent (*Drosophila melanogaster*) | $Ir48b^1$:$w^{1118}$;Mi{GFP$^{E.3xP3}$=ET1}$Ir48b^{MB02315}$ | Bloomington *Drosophila* Stock Center | BDSC:23473 | |
| Genetic reagent (*Drosophila melanogaster*) | $Ir51b^1$:$w^{1118}$;PBac{$w^{+mC}$=PB}row$^{c00387}$ $Ir51b^{c00387}$ | Bloomington *Drosophila* Stock Center | BDSC:10046 | |
| Genetic reagent (*Drosophila melanogaster*) | $Ir52a^1$ | **Rimal et al., 2019** | | Provided by Dr. Y. Lee |
| Genetic reagent (*Drosophila melanogaster*) | $Ir52b^1$:$w^{1118}$;Mi{GFP$^{E.3xP3}$=ET1}$Ir52b^{MB02231}$/SM6a | Bloomington *Drosophila* Stock center | BDSC:25212 | |
| Genetic reagent (*Drosophila melanogaster*) | $Ir52c^1$:$w^{1118}$; Mi{GFP$^{E.3xP3}$=ET1}$Ir52c^{MB04402}$ | Bloomington *Drosophila* Stock center | BDSC:24580 | |
| Genetic reagent (*Drosophila melanogaster*) | $Ir56a^1$ | **Rimal et al., 2019** | | Provided by Dr. Y. Lee |
| Genetic reagent (*Drosophila melanogaster*) | $Ir56b^1$:$w^{1118}$;Mi{GFP$^{E.3xP3}$=ET1}$Ir56b^{MB09950}$ | Bloomington *Drosophila* Stock Center | BDSC:27818 | |
| Genetic reagent (*Drosophila melanogaster*) | $Ir56d^1$:w[*];$Ir56d^1$ | Bloomington *Drosophila* Stock Center | BDSC:81249 | |

*Continued on next page*

*Continued*

| Reagent type (species) or resource | Designation | Source or reference | Identifiers | Additional information |
|---|---|---|---|---|
| Genetic reagent (*Drosophila melanogaster*) | *Ir60b³* | **Sang et al., 2024** | | Provided by Dr. Y. Lee |
| Genetic reagent (*Drosophila melanogaster*) | *Ir62a¹*:y¹w*;Mi{y^{+mDint2}=MIC}Ir62a^{MI00895} Iml1^{MI00895}/TM3, Sb¹ Ser¹ | Bloomington *Drosophila* Stock Center | BDSC:32713 | |
| Genetic reagent (*Drosophila melanogaster*) | *Ir67a¹*: y¹ w*; Mi{y^{+mDint2}=MIC}Ir67a^{MI11288} | Bloomington *Drosophila* Stock Center | BDSC:56583 | |
| Genetic reagent (*Drosophila melanogaster*) | *Ir75d¹*:w^{1118};Mi{GFP^{E.3xP3}=ET1}Ir75d^{MB04616} | Bloomington *Drosophila* Stock Center | BDSC:24205 | |
| Genetic reagent (*Drosophila melanogaster*) | *Ir76b¹* | **Zhang et al., 2013a** | | Provided by Dr. C. Montell |
| Genetic reagent (*Drosophila melanogaster*) | *Ir85a¹*:w^{1118};Mi{GFP^{E.3xP3}=ET1}Ir85a^{MB04613} Pif1A^{MB04613} | Bloomington *Drosophila* Stock Center | BDSC:24590 | |
| Genetic reagent (*Drosophila melanogaster*) | *Ir92a¹*:w^{1118};Mi{GFP^{E.3xP3}=ET1}Ir92a^{MB03705} | Bloomington *Drosophila* Stock Center | BDSC:23638 | |
| Genetic reagent (*Drosophila melanogaster*) | *Ir94a¹* | **Rimal et al., 2019** | | Provided by Dr. Y. Lee |
| Genetic reagent (*Drosophila melanogaster*) | *Ir94b¹*; Mi{GFP^{E.3xP3}=ET1}Ir94b^{MB02190} | Bloomington *Drosophila* Stock Center | BDSC:23424 | |
| Genetic reagent (*Drosophila melanogaster*) | *Ir94c¹* | **Rimal et al., 2019** | | Provided by Dr. Y. Lee |
| Genetic reagent (*Drosophila melanogaster*) | *Ir94d¹*:y¹w[;Mi{y^{+mDint2}=MIC} Ir94d^{MI01659}CG17380^{MI01659} | Bloomington *Drosophila* Stock Center | BDSC:33132 | |
| Genetic reagent (*Drosophila melanogaster*) | *Ir94f¹*: y¹ w*; Mi{y^{+mDint2}=MIC}Ir94f^{MI00928} | Bloomington *Drosophila* Stock Center | BDSC:33095 | |
| Genetic reagent (*Drosophila melanogaster*) | *Ir94g¹*: w^{1118}; Mi{GFP^{E.3xP3}=ET1}Ir94g^{MB07445} | Bloomington *Drosophila* Stock Center | BDSC:25551 | |
| Genetic reagent (*Drosophila melanogaster*) | *Ir94h¹* | **Rimal et al., 2019** | | Provided by Dr. Y. Lee |
| Genetic reagent (*Drosophila melanogaster*) | *Ir100a¹*: w^{1118};P{w^{+mC}=EP}Ir100a^{G19846} CG42233^{G19846} | Bloomington *Drosophila* Stock Center | BDSC:31853 | |
| Genetic reagent (*Drosophila melanogaster*) | *UAS-Ir25a* | **Lee et al., 2018** | | Provided by Dr. Y. Lee |
| Genetic reagent (*Drosophila melanogaster*) | *UAS-Ir51b* | **Dhakal et al., 2021** | | Provided by Dr. Y. Lee |

*Continued on next page*

*Continued*

| Reagent type (species) or resource | Designation | Source or reference | Identifiers | Additional information |
|---|---|---|---|---|
| Genetic reagent (*Drosophila melanogaster*) | Gr33a[1] | *Moon et al., 2009* | | Provided by Dr. C. Montell |
| Genetic reagent (*Drosophila melanogaster*) | Gr33a-GAL4 | *Moon et al., 2009* | | Provided by Dr. C. Montell |
| Genetic reagent (*Drosophila melanogaster*) | Gr47a[1] | *Lee et al., 2015* | | Provided by Dr. C. Montell |
| Genetic reagent (*Drosophila melanogaster*) | elav-GAL4;UAS-Dicer2 | Bloomington *Drosophila* Stock Center | BDSC:25750 | |
| Genetic reagent (*Drosophila melanogaster*) | Gr39a[1] | Bloomington *Drosophila* Stock Center | BDSC:10562 | |
| Genetic reagent (*Drosophila melanogaster*) | Gr93a[3] | *Lee et al., 2009* | | Provided by Dr. Y. Lee |
| Genetic reagent (*Drosophila melanogaster*) | UAS-Kir2.1 | Bloomington *Drosophila* Stock Center | BDSC:6596 | |
| Genetic reagent (*Drosophila melanogaster*) | ΔGr32a | *Miyamoto and Amrein, 2008* | | Provided by Dr. H. Amrein |
| Genetic reagent (*Drosophila melanogaster*) | Gr66a[ex83] | *Moon et al., 2006* | | Provided by Dr. C. Montell |
| Genetic reagent (*Drosophila melanogaster*) | Gr89a[1] | Korea *Drosophila* Resource Center | KDRC: (*Sung et al., 2017*) | |
| Genetic reagent (*Drosophila melanogaster*) | Ir7c[GAL4] | *McDowell et al., 2022* | | Provided by Dr. M. Gordon |
| Genetic reagent (*Drosophila melanogaster*) | Ir20a[1] | *Ganguly et al., 2017* | | Provided by Dr. A. Dahanukar |
| Genetic reagent (*Drosophila melanogaster*) | Ir25a-GAL4 | *Benton et al., 2009* | | Provided by Dr. L. Vosshall |
| Genetic reagent (*Drosophila melanogaster*) | UAS-Ir76b | *Moon et al., 2006* | | Provided by Dr. C. Montell |
| Genetic reagent (*Drosophila melanogaster*) | Ir76b-GAL4 | *Moon et al., 2006* | | Provided by Dr. C. Montell |
| Genetic reagent (*Drosophila melanogaster*) | ppk23-GAL4 | *Thistle et al., 2012* | | Provided by Dr. K. Scott |
| Genetic reagent (*Drosophila melanogaster*) | ppk28-GAL4 | *Cameron et al., 2010* | | Provided by Dr. H. Amrein |

*Continued on next page*

*Continued*

| Reagent type (species) or resource | Designation | Source or reference | Identifiers | Additional information |
|---|---|---|---|---|
| Genetic reagent (*Drosophila melanogaster*) | Gr5a-GAL4 | **Dahanukar et al., 2001** | | Provided by Dr. H. Amrein |
| Genetic reagent (*Drosophila melanogaster*) | UAS-Kir2.1 | Bloomington *Drosophila* Stock Center | BDSC:6595 | |
| Genetic reagent (*Drosophila melanogaster*) | Ir7g[2] | **Pradhan et al., 2024** | | Provided by Dr. Y. Lee |
| Genetic reagent (*Drosophila melanogaster*) | UAS-Ir7g | **Pradhan et al., 2024** | | Provided by Dr. Y. Lee |
| Genetic reagent (*Drosophila melanogaster*) | UAS-Ir56d | **Sánchez-Alcañiz et al., 2018** | | Provided by Dr. R. Benton |
| Genetic reagent (*Drosophila melanogaster*) | Ir56d-GAL4 | Korea *Drosophila* Resource Center | KDRC:2307 | |
| Genetic reagent (*Drosophila melanogaster*) | Ir56d[2] | Bloomington *Drosophila* Stock Center | BDSC:81250 | |
| Genetic reagent (*Drosophila melanogaster*) | Ir51b[2] | **Dhakal et al., 2021** | | Provided by Dr. Y. Lee |
| Genetic reagent (*Drosophila melanogaster*) | BC/CyO;Gr66a-I-GFP,UAS-dsred/TM6b | **Weiss et al., 2011** | | Provided by Dr. J.R. Carlson |
| Genetic reagent (*Drosophila melanogaster*) | Ir7g RNAi | Vienna *Drosophila* Resource Center | VDRC:100885 | |
| Genetic reagent (*Drosophila melanogaster*) | Ir25a RNAi | Vienna *Drosophila* Resource Center | VDRC:106731 | |
| Genetic reagent (*Drosophila melanogaster*) | Ir51b RNAi | Vienna *Drosophila* Resource Center | VDRC:29984 | |
| Genetic reagent (*Drosophila melanogaster*) | Ir56d RNAi | Vienna *Drosophila* Resource Center | VDRC6112 | |
| Genetic reagent (*Drosophila melanogaster*) | Ir76b RNAi | Vienna *Drosophila* Resource Center | VDRC8433 | |
| Genetic reagent (*Drosophila melanogaster*) | trpA1[1] | **Kwon et al., 2008** | | Provided by Dr. C. Montell |
| Genetic reagent (*Drosophila melanogaster*) | trpl[29134] | **Niemeyer et al., 1996** | | Provided by Dr. C. Montell |
| Genetic reagent (*Drosophila melanogaster*) | trpγ[1] | **Akitake et al., 2015** | | Provided by Dr. C. Montell |

*Continued*

| Reagent type (species) or resource | Designation | Source or reference | Identifiers | Additional information |
|---|---|---|---|---|
| Genetic reagent (*Drosophila melanogaster*) | *amo[1]* | **Watnick et al., 2003** | | Provided by Dr. C. Montell |
| Genetic reagent (*Drosophila melanogaster*) | *iav[3621]* | Bloomington *Drosophila* Stock center | BDSC:24768 | |
| Genetic reagent (*Drosophila melanogaster*) | *nan[36a]* | **Kim et al., 2003** | | Provided by Dr. C. Kim |
| Genetic reagent (*Drosophila melanogaster*) | *trp[343]* | **Tracey et al., 2003** | | Provided by Dr. C. Montell |
| Genetic reagent (*Drosophila melanogaster*) | *pyx[3]* | **Lee et al., 2005** | | Provided by Dr. Y. Lee |
| Genetic reagent (*Drosophila melanogaster*) | *wtrw[ex]* | **Kim et al., 2010** | | Provided by Dr. C. Montell |
| Genetic reagent (*Drosophila melanogaster*) | *pain[2]* | **Tracey et al., 2003** | | Provided by Dr. S. Benzer |
| Antibody | Rabbit anti-DsRed(rabbit polyclonal) | Takara | Cat # 632496 RRID:AB_10013483 | 1:1000 (1 µL) |
| Antibody | Goat anti-mouse Alexa Fluor 568 | Thermo fisher/Invitrogen | Cat # A11004 RRID:AB_2534072 | 1:200 (1 µL) |
| Antibody | Mouse anti-GFP (mouse monoclonal) | Molecular probe | Cat # A11120 RRID:AB_221568 | 1:1000 (1 µL) |
| Antibody | Goat anti-mouse Alexa Fluor 488 | Thermo Fisher/Invitrogen | Cat # A11029 RRID:AB_2534088 | 1:200 (1 µL) |
| Chemical compound or drug | Cholesterol | Sigma-Aldrich Co. | Cat# C4951 | |
| Chemical compound or drug | Sucrose | Sigma-Aldrich Co. | Cat# S9378 | |
| Chemical compound or drug | Tricholine citrate | Sigma-Aldrich Co. | Cat# T0252 | |
| Chemical compound or drug | Stigmasterol | Sigma-Aldrich Co. | Cat# S2424 | |
| Chemical compound or drug | Sulforhodamine B | Sigma-Aldrich Co. | Cat# 230162 | |
| Chemical compound or drug | Brilliant blue FCF | Wako Pure Chemical Industry Ltd. | Cat# 027–12842 | |
| Chemical compound or drug | Methyl beta cyclodextrin | Sigma-Aldrich Co. | Cat# 332615 | |
| Chemical compound or drug | Paraformaldehyde | Electron Microscopy Sciences | Cat # 15710 | 1:500 Provided by Dr. J.A. Veenstra |
| Chemical compound or drug | Goat Serum, New Zealand origin | Gibco | Cat # 16210064 | |
| Software, algorithm | *Origin Pro Version* | OriginLab corporation | RRID:SCR_002815 | https://www.originlab.com/ |
| Software, algorithm | *Graphpad Prism* | GraphPad | RRID:SCR_002798 | https://www.graphpd.com/ |
| Software, algorithm | Autospike 3.1 software | | | https://www.syntech.co.za/ |

## Chemical reagents

The following chemicals and reagents were purchased from Sigma-Aldrich: cholesterol (catalog no. C4951), MβCD (catalog no. 332615), stigmasterol (catalog no. S2424), propionic acid (catalog no. 402907), butyric acid (catalog no. B103500), acetic acid (catalog no. A8976), caffeine (catalog no. C0750), denatonium benzoate (catalog no. D5765), sulforhodamine B (catalog no. 230162), tricholine citrate (TCC; catalog no. T0252), umbelliferone (catalog no. 24003), berberine sulfate hydrate (catalog no. B0451), cholesteroloroquine diphosphate salt (catalog no. C6628), lobeline hydrocholesteroloride (catalog no. 141879), quinine hydrocholesteroloride dihydrate (catalog no. Q1125), papaverine hydro-cholesteroloride (catalog no. P3510), strychnine hydrocholesteroloride (catalog no. S8753), coumarin (catalog no. C4261), and sucrose (catalog no. S9378). Brilliant Blue FCF (catalog no. 027–12842) was purchased from Wako Pure Chemical Industry Ltd. The following antibodies were purchased from the following sources: mouse anti-GFP antibody (Molecular Probes, catalog no. A11120), rabbit anti-DsRed (TaKaRa Bio, catalog no. 632496), goat anti-mouse Alexa Fluor 488 (Thermo Fisher Scientific, catalog no. A11029), and goat anti-rabbit Alexa Fluor 568 (catalog no. A11011, Thermo Fisher Scientific/Invitrogen).

## Binary food choice assay

In accordance with a previous study, we carried out experiments involving binary food choice tests (*Aryal et al., 2022a*). To initiate each experiment, a group of 50–70 flies (aged 3–6 d, consisting of both males and females) were subjected to an 18 hr period of fasting in a controlled humidity chamber. The subsequent procedures included the preparation of two distinct food sources, both incorporating 1% agarose as the base. The first food source was enriched with 2 mM sucrose, and the second source contained different concentrations of cholesterol in addition to 2 mM sucrose. To distinguish between these two food sources, we introduced blue food coloring dye (0.125 mg/mL brilliant blue FCF) to one and red food coloring dye (0.1 mg/mL sulforhodamine B) to the other. We evenly distributed these prepared solutions into the wells of a 72-well microtiter dish (Thermo Fisher Scientific, catalog no. 438733), alternating between the two options. Approximately 50–70 starved flies were introduced to the plate within approximately 30 min of food preparation. The flies were allowed to feed at room temperature (25 °C) for 90 min, which occurred in a dark, humid environment to maintain consistent conditions. Afterward, the tested flies were carefully frozen at −20 °C for further analysis. With the aid of a stereomicroscope, we observed and categorized the colors of their abdomens as either blue ($N_B$), red ($N_R$), or purple ($N_P$). For each fly, we calculated the PI, a value derived from the combinations of dye and tastant, as follows: $(N_B - N_R)/(N_R + N_B + _{NP})$ or $(N_R - N_B)/(N_R + N_B + _{NP})$. A PI of either 1.0 or −1.0 indicated a complete preference for one of the food alternatives, and a PI of 0.0 signified no bias among the flies toward either option.

## Tip recording assay

The tip recording assay was conducted according to previously established protocols (*Moon et al., 2006*; *Shrestha et al., 2022*). Flies of both sexes, aged between 4 and 7 d, were gently anesthetized on a bed of ice . A reference glass electrode containing Ringer's solution was inserted into the thoracic region of the flies. Subsequently, the electrode was incrementally advanced toward the proboscis of each fly. This precise process was repeated over multiple days to ensure the reliability and consistency of the results. To stimulate the sensilla, a recording pipette with a tip diameter ranging from 10 to 20 µm was connected to a preamplifier. The pipette was filled with a blend of chemical stimulants dissolved in a 30 mM TCC solution, which served as the electrolyte solution. Signal amplification was achieved through a Syntech signal connection interface box and a band-pass filter spanning a range of 100–3000 Hz. These amplified signals were recorded at a sampling rate of 12 kHz and subsequently analyzed using AutoSpike 3.1 software (Syntech). To ensure the integrity of the recorded signals, all recordings were carried out at regular 1 min intervals.

## Immunohistochemistry

Immunohistochemistry analysis was performed following established procedures (*Lee et al., 2012*). The labellum or brain of the flies were dissected and fixed using a 4% paraformaldehyde solution (Electron Microscopy Sciences, catalog no. 15710) in PBS-T (1 X phosphate-buffered saline containing 0.2% Triton X-100) for 25 min at 4 °C. The fixed tissues were thoroughly rinsed three times with

PBS-T for 15 min each, precisely bisected using a razor blade, and then incubated for 30 min at room temperature in a blocking buffer composed of 0.5% goat serum in 1 X PBS-T. Primary antibodies (1:1000 dilution; mouse anti-GFP [Molecular Probes, catalog no. A11120] and rabbit anti-DsRed [TaKaRa Bio, catalog no. 632496]) were added to freshly prepared blocking buffer and incubated with the samples overnight at 4 °C. After overnight incubation, the samples underwent an additional round of thorough washing with PBS-T at 4 °C before being exposed to secondary antibodies (1:200 dilution in blocking buffer; goat anti-mouse Alexa Fluor 488 [Thermo Fisher Scientific, catalog no. A11029] and goat anti-rabbit Alexa Fluor 568 [Thermo Fisher Scientific/Invitrogen, catalog no. A11011]) for 4 hr at 4 °C. After another three rounds of washing with PBS-T, the tissues were immersed in 1.25 X PDA mounting buffer (37.5% glycerol, 187.5 mM NaCl, and 62.5 mM Tris, pH 8.8) and examined using an inverted Leica LASX confocal microscope for visualization and analysis.

## Quantification and statistical analyses

We processed and conducted data analysis using GraphPad Prism version 8.0 (RRID:SCR 002798). Each experiment was independently replicated on different days, and the numbers of trials for each experiment are indicated as data points on the graphs. Error bars on the graphs represent the standard error of the mean (SEM). Single-factor ANOVA was combined with Scheffe's post hoc analysis to compare multiple datasets. All statistical analyses were carried out using Origin (Origin Lab Corporation, RRID:SCR 002815). In the figures, asterisks are used to indicate statistical significance, with denotations of $*p<0.05$ and $**p<0.01$.

## Acknowledgements

This work was supported by grants to YL from the National Research Foundation of Korea (NRF) funded by the Ministry of Science and ICT (RS-2021-NR058319) and the Korea Environmental Industry and Technology Institute (KEITI) grant funded by the Ministry of Environment of Korea. RNP was supported by the Global Scholarship Program for Foreign Graduate Students at Kookmin University in Korea. CM is supported by grants from the National Institute on Deafness and other Communication Disorders (DC007864 and DC016278).

## Additional information

### Funding

| Funder | Grant reference number | Author |
|---|---|---|
| National Research Foundation of Korea | RS-2021-NR058319 | Youngseok Lee |
| National Institute on Deafness and Other Communication Disorders | DC007864 | Craig Montell |
| National Institute on Deafness and Other Communication Disorders | DC016278 | Craig Montell |

The funders had no role in study design, data collection and interpretation, or the decision to submit the work for publication.

### Author contributions

Roshani Nhuchhen Pradhan, Conceptualization, Data curation, Validation, Investigation, Visualization, Writing – original draft; Craig Montell, Funding acquisition, Writing – review and editing; Youngseok Lee, Conceptualization, Supervision, Funding acquisition, Writing – review and editing

### Author ORCIDs

Roshani Nhuchhen Pradhan (iD) https://orcid.org/0000-0001-6772-6357
Craig Montell (iD) https://orcid.org/0000-0001-5637-1482
Youngseok Lee (iD) https://orcid.org/0000-0003-0459-1138

Reviewer #1 (Public review): https://doi.org/10.7554/eLife.106256.2.sa1
Reviewer #2 (Public review): https://doi.org/10.7554/eLife.106256.2.sa2
Reviewer #3 (Public review): https://doi.org/10.7554/eLife.106256.2.sa3
Author response https://doi.org/10.7554/eLife.106256.2.sa4

## Additional files

### Supplementary files
Supplementary file 1. Electrophysiological analysis and binary food choice assay of MβCD, cholesterol, and stigmasterol, and immunohistochemical analysis of Ir56d-GAL4 co-localization with a bitter GRN reporter were performed.

MDAR checklist

### Data availability
Source data for all figures contained in the manuscript have been deposited in 'figshare' (https://doi.org/10.6084/m9.figshare.28293062).

The following previously published dataset was used:

| Author(s) | Year | Dataset title | Dataset URL | Database and Identifier |
|---|---|---|---|---|
| Pradhan RN, Montell C, Lee Y | 2025 | Cholesterol taste avoidance in *Drosophila melanogaster* | https://doi.org/10.6084/m9.figshare.28293062 | figshare, 10.6084/m9.figshare.28293062 |

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
