## [Editor Report · eLife Assessment]

This **useful** study provides **convincing** evidence that *Drosophila* can taste cholesterol through a subset of bitter-sensing gustatory receptor neurons, and that flies avoid high-cholesterol food. However, the same receptors have been previously found to be involved in the detection of multiple seemingly unrelated chemicals, and the reported expression patterns of these receptors contradict past reports. These caveats are not mentioned in the paper, raising critical concerns about the study's conclusions.

---

## [Referee Report · Reviewer #1 (Public review)]

Summary:

Pradhan et al investigated the potential gustatory mechanisms that allow flies to detect cholesterol. They found that flies are indifferent to low cholesterol and avoid high cholesterol. They further showed that the ionotropic receptors Ir7g, Ir51b, and Ir56d are important for the cholesterol sensitivity in bitter neurons. The figures are clear and the behavior result is interesting. However, I have several major comments, especially on the discrepancy of the expression of these Irs with other lab published results, and the confusing finding that the same receptors (Ir7g, Ir51b) have been implicated in the detection of various seemingly unrelated compounds.

Strengths:

The results are very well presented, the figures are clear and well-made, text is easy to follow.

Weaknesses:

(1) Regarding the expression of Ir56d. The reported Ir56d expression pattern contradicts multiple previous studies (Brown et al., 2021 eLife, Figure 6a-c; Sanchez-Alcaniz et al., 2017 Nature Communications, Figure 4e-h; Koh et al., 2014 Neuron, Figure 3b). These studies, using three different driver lines, consistently showed Ir56d expression in sweet-sensing neurons and taste peg neurons. Importantly, Sanchez-Alcaniz et al. demonstrated that Ir56d is not expressed in Gr66a-expressing (bitter) neurons. This discrepancy is critical since Ir56d is identified as the key subunit for cholesterol detection in bitter neurons, and misexpression of Ir7g and Ir51b together is insufficient to confer cholesterol sensitivity (Fig.4b,d). Which Ir56d-GAL4 (and Gr66a-I-GFP) line was used in this study? Is there additional evidence (scRNA sequencing, in-situ hybridization, or immunostaining) supporting Ir56d expression in bitter neurons?

(2) Ir51b has previously been implicated in detecting nitrogenous waste (Dhakal 2021), lactic acid (Pradhan 2024), and amino acids (Aryal 2022), all by the same lab. Additionally, both Ir7g and Ir51b have been implicated in detecting cantharidin, an insect-secreted compound that flies may or may not encounter in the wild, by the same lab. Is Ir51b proposed to be a specific receptor for these chemically distinct compounds or a general multimodal receptor for aversive stimuli? Unlike other multimodal bitter receptors, the expression level of Ir51b is rather low and it's unclear which subset of GRNs express this receptor. The chemical diversity among nitrogenous waste, amino acids, lactic acid, cantharidin, and cholesterol raises questions about the specificity of these receptors and warrants further investigation and at a minimum discussion in this paper. Given the wide and seemingly unrelated sensitivity of Ir51b and Ir7g to these compounds I'm leaning towards the hypothesis that at least some of these is non-specific and ecologically irrelevant without further supporting evidence from the authors.

(3) The Benton lab Ir7g-GAL4 reporter shows no expression in adults. Additionally, two independent labellar RNA sequencing studies (Dweck, 2021 eLife; Bontonou et al., 2024 Nature Communications) failed to detect Ir7g expression in the labellum. This contradicts the authors' previous RT-PCR results (Pradhan 2024 Fig. S4, Journal of Hazardous Materials) showing Ir7g expression in the labellum. Additionally the Benton and Carlson lab Ir51b-GAL4 reporters show no expression in adults as well. Please address these inconsistencies.

(4) The premise that high cholesterol intake is harmful to flies, which makes sensory mechanisms for cholesterol avoidance necessary, is interesting but underdeveloped. Animal sensory systems typically evolve to detect ecologically relevant stimuli with dynamic ranges matching environmental conditions. Given that *Drosophila* primarily consume fruits and plant matter (which contain minimal cholesterol) rather than animal-derived foods (which contain higher cholesterol), the ecological relevance of cholesterol detection requires more thorough discussion. Furthermore, at high concentrations, chemicals often activate multiple receptors beyond those specifically evolved for their detection. If the cholesterol concentrations used in this study substantially exceed those encountered in the fly's natural diet, the observed responses may represent an epiphenomenon rather than an ecologically and ethologically relevant sensory mechanism. What is the cholesterol content in flies' diet and how does that compare to the concentrations used in this paper?

---

## [Referee Report · Reviewer #2 (Public review)]

Summary:

In Cholesterol Taste Avoidance in *Drosophila melanogaster*, Pradhan et al. used behavioral and electrophysiological assays to demonstrate that flies can: (1) detect cholesterol through a subset of bitter-sensing gustatory receptor neurons (GRNs) and (2) avoid consuming food with high cholesterol levels. Mechanistically, they identified five members of the IR family as necessary for cholesterol detection in GRNs and for the corresponding avoidance behavior. Ectopic expression experiments further suggested that Ir7g + Ir56d or Ir51b + Ir56d may function as tuning receptors for cholesterol detection, together with the Ir25a and Ir76b co-receptors.

Strengths:

The experimental design of this study was logical and straightforward. Leveraging their expertise in the *Drosophila* taste system, the research team identified the molecular and cellular basis of a previously unrecognized taste category, expanding our understanding of gustation. A key strength of the study was its combination of electrophysiological recordings with behavioral genetic experiments.

Weaknesses:

My primary concern with this study is the lack of a systematic survey of the IRs of interest in the labellum GRNs. Consequently, there is no direct evidence linking the expression of putative cholesterol IRs to the B GRNs in the S6 and S7 sensilla.

Specifically, the authors need to demonstrate that the IR expression pattern explains cholesterol sensitivity in the B GRNs of S6 and S7 sensilla, but not in other sensilla. Instead of providing direct IR expression data for all candidate IRs (as shown for Ir56d in Figure 2-figure supplement 1F), the authors rely on citations from several studies (Lee, Poudel et al. 2018; Dhakal, Sang et al. 2021; Pradhan, Shrestha et al. 2024) to support their claim that Ir7g, Ir25a, Ir51b, and Ir76b are expressed in B GRNs (Lines 192-194). However, none of these studies provide GAL4 expression or in situ hybridization data to substantiate this claim.

Without a comprehensive IR expression profile for GRNs across all taste sensilla, it is difficult to interpret the ectopic expression results observed in the B GRN of the I9 sensillum or the A GRN of the L-sensillum (Figure 4). It remains equally plausible that other tuning IRs-beyond the co-receptor Ir25a and Ir76b-could interact with the ectopically expressed IRs to confer cholesterol sensitivity, rather than the proposed Ir7g + Ir56d or Ir51b + Ir56d combinations.

---

## [Referee Report · Reviewer #3 (Public review)]

Summary:

Whether and how animals can taste cholesterol is not well understood. The study provides evidence that (1) cholesterol activates a subset of bitter-sensing gustatory receptor neurons (GRNs) in the fly labellum, but not other types of GRNs, (2) flies show aversion to high concentrations of cholesterol, and this is mediated by bitter GRNs, and (3) cholesterol avoidance depends on a specific set of ionotropic receptor (IR) subunits acting in bitter GRNs. The claims of the study are supported by electrophysiological recordings, genetic manipulations, and behavioral readouts.

Strengths:

Cholesterol taste has not been well studied, and the paper provides new insight into this question. The authors took a comprehensive and rigorous approach in several different parts of the paper, including screening the responses of all 31 labellar sensilla, screening a large panel of receptor mutants, and performing misexpression experiments with nearly every combination of the 5 IRs identified. The effects of the genetic manipulations are very clear and the results of electrophysiological and behavioral studies match nicely, for the most part. The appropriate controls are performed for all genetic manipulations.

Weaknesses:

The weaknesses of the study, described below, are relatively minor and do not detract from the main conclusions of the paper.

(1) The paper does not state what concentrations of cholesterol are present in *Drosophila*'s natural food sources. Are the authors testing concentrations that are ethologically relevant?

(2) The paper does not state or show whether the expression of IR7g, IR51b, and IR56d is confined to bitter GRNs. Bitter-specific expression of at least some of these receptors would be necessary to explain why bitter GRNs but not sugar GRNs (or other GRN types) normally show cholesterol responses.

(3) The authors only investigated the responses of GRNs in the labellum, but GRN responses in the leg may also contribute to the avoidance of cholesterol feeding. Alternatively, leg GRNs might contribute to cholesterol attraction that is unmasked when bitter GRNs are silenced. In support of this possibility, Ahn et al. (2017) showed that Ir56d functions in sugar GRNs of the leg to promote appetitive responses to fatty acids.

(4) The authors might consider using proboscis extension as an additional readout of taste attraction or aversion, which would help them more directly link the labellar GRN responses to a behavioral readout. Using food ingestion as a readout can conflate the contribution of taste with post-ingestive effects, and the regulation of food ingestion also may involve contributions from GRNs on multiple organs, whereas organ-specific contributions can be dissociated using proboscis extension. For example, does presenting cholesterol on the proboscis lead to aversive responses in the proboscis extension assay (e.g., suppression of responses to sugar)? Does this aversion switch to attraction when bitter GRNs are silenced, as with the feeding assay?

(5) The authors claim that the cholesterol receptor is composed of IR25a, IR76b, IR56d, and either IR7g or IR51b. While the authors have shown that IR25a and IR76b are each required for cholesterol sensing, they did not show that both are required components of the same receptor complex. If the authors are relying on previous studies to make this assumption, they should state this more clearly. Otherwise, I think further misexpression experiments may be needed where only IR25a or IR76b, but not both, are expressed in GRNs.

---

## [Author Response]

**Public Reviews:**

**Reviewer #1 (Public review):**
Summary:Pradhan et al investigated the potential gustatory mechanisms that allow flies to detect cholesterol. They found that flies are indifferent to low cholesterol and avoid high cholesterol. They further showed that the ionotropic receptors Ir7g, Ir51b, and Ir56d are important for the cholesterol sensitivity in bitter neurons. The figures are clear and the behavior result is interesting. However, I have several major comments, especially on the discrepancy of the expression of these Irs with other lab published results, and the confusing finding that the same receptors (Ir7g, Ir51b) have been implicated in the detection of various seemingly unrelated compounds.Strengths:The results are very well presented, the figures are clear and well-made, text is easy to follow.Weaknesses:(1) Regarding the expression of Ir56d. The reported Ir56d expression pattern contradicts multiple previous studies (Brown et al., 2021 eLife, Figure 6a-c; Sanchez-Alcaniz et al., 2017 Nature Communications, Figure 4e-h; Koh et al., 2014 Neuron, Figure 3b). These studies, using three different driver lines, consistently showed Ir56d expression in sweet-sensing neurons and taste peg neurons. Importantly, Sanchez-Alcaniz et al. demonstrated that Ir56d is not expressed in Gr66a-expressing (bitter) neurons. This discrepancy is critical since Ir56d is identified as the key subunit for cholesterol detection in bitter neurons, and misexpression of Ir7g and Ir51b together is insufficient to confer cholesterol sensitivity (Fig.4b,d). Which Ir56d-GAL4 (and Gr66a-I-GFP) line was used in this study? Is there additional evidence (scRNA sequencing, in-situ hybridization, or immunostaining) supporting Ir56d expression in bitter neurons?

We agree that the expression pattern of *Ir56d* diverges from two prior reports . The studies by Brown et al. and Koh et al. employed the same *Ir56d-GAL4* driver line, which exhibited expression in sweet-sensing gustatory receptor neurons (GRNs) and taste peg neurons, but not bitter GRNs (the Sanchez-Alcaniz et al. paper did not use an *Ir56d-Gal4*).

In our study, we used a *Ir56d-GAL4* driver line (KDRC:2307) and the *Gr66a-I-GFP* reporter line (Weiss et al., 2011 Neuron). This is a crucial distinction, as differences in the regulatory regions used to generate different driver lines are well known to underlie differences in expression patterns. Our double-labeling experiments revealed co-expression of *Ir56d* with *Gr66a*-positive bitter GRNs specifically within the S6 and S7 sensilla—types previously shown to exhibit strong electrophysiological responses to cholesterol (Figure 2—figure supplement 1F).

We believe this observation is biologically significant and consistent with our functional data. Specifically, targeted expression of *Ir56d* in bitter neurons using the *Gr33a-GAL4* was sufficient to rescue cholesterol avoidance behavior in *Ir56d1* mutants (Figure 3G). These results demonstrate that *Ir56d* plays a functional role in bitter GRNs for cholesterol detection. The convergence of genetic, behavioral, and electrophysiological data presented in our study provides compelling support for this previously unappreciated expression pattern and function of *Ir56d*.

(2) Ir51b has previously been implicated in detecting nitrogenous waste (Dhakal 2021), lactic acid (Pradhan 2024), and amino acids (Aryal 2022), all by the same lab. Additionally, both Ir7g and Ir51b have been implicated in detecting cantharidin, an insect-secreted compound that flies may or may not encounter in the wild, by the same lab. Is Ir51b proposed to be a specific receptor for these chemically distinct compounds or a general multimodal receptor for aversive stimuli? Unlike other multimodal bitter receptors, the expression level of Ir51b is rather low and it's unclear which subset of GRNs express this receptor. The chemical diversity among nitrogenous waste, amino acids, lactic acid, cantharidin, and cholesterol raises questions about the specificity of these receptors and warrants further investigation and at a minimum discussion in this paper. Given the wide and seemingly unrelated sensitivity of Ir51b and Ir7g to these compounds I'm leaning towards the hypothesis that at least some of these is non-specific and ecologically irrelevant without further supporting evidence from the authors.

While it is true that IR51b and IR7g are responsive to a range of compounds, they share chemical features such as nitrogen-containing groups, hydrophobicity, or amphipathic structures suggesting that recognition of these chemicals may be mediated by the same or overlapping domains within the receptor complexes. These features could facilitate binding to a structurally diverse yet chemically related groups of aversive ligands.

In the case of cholesterol, while its sterol ring system is distinct from the other compounds, it shares hydrophobic and amphipathic properties that may enable interaction with these receptors via similar structural motifs. Importantly, our data demonstrate that *Ir51b* and *Ir7g* are necessary but not sufficient on their own to confer cholesterol sensitivity, indicating that additional co-factors or receptor subunits are required for full functionality (Figure 4B, D). Furthermore, our dose-response analysis (Figure 3F) shows that *Ir7g* is particularly important at higher cholesterol concentrations, supporting the idea of graded sensitivity rather than indiscriminate activation. This suggests that these receptors may have evolved to recognize cholesterol and its analogs (e.g., phytosterols such as stigmasterol, yet to be tested), which are naturally found in the fly’s diet (e.g., yeast and plant-derived matter), as ecologically relevant cues signaling microbial contamination, lipid imbalance, or dietary overconsumption.

We acknowledge the reviewer’s concern regarding the relatively low expression levels of *Ir51b* and *Ir7g*. However, we note that low transcript abundance does not necessarily equate to diminished physiological relevance. Finally, we agree that the chemical diversity of ligands associated with *Ir51b* and *Ir7g* warrants deeper investigation, particularly through structure-function studies aimed at identifying ligand-binding domains and receptor-ligand interactions at atomic resolution.

(3) The Benton lab Ir7g-GAL4 reporter shows no expression in adults. Additionally, two independent labellar RNA sequencing studies (Dweck, 2021 eLife; Bontonou et al., 2024 Nature Communications) failed to detect Ir7g expression in the labellum. This contradicts the authors' previous RT-PCR results (Pradhan 2024 Fig. S4, Journal of Hazardous Materials) showing Ir7g expression in the labellum. Additionally the Benton and Carlson lab Ir51b-GAL4 reporters show no expression in adults as well. Please address these inconsistencies.

With respect to *Ir7g*, we acknowledge that the *Ir7g-GAL4* reporter line from the Benton lab does not exhibit detectable expression in adult labella. Furthermore, two independent transcriptomic studies—Dweck et al., 2021 (eLife) and Bontonou et al., 2024 (Nature Communications) also did not detect *Ir7g* transcripts in bulk RNA-seq datasets derived from adult labella. However, our previously published RT-PCR data (Pradhan et al., 2024, Journal of Hazardous Materials, Fig. S4) revealed *Ir7g* expression in labellar tissue, albeit at low levels. Our RT-PCR includes an internal control (*tubulin*) with the same reaction tube with control and the *Ir7g* mutant as a negative control. Therefore, we stand behind the findings that *Ir7g* is expressed in the labellum.

We would like to point out that RT-PCR is more sensitive and better-suited to detect low-abundance transcripts than bulk RNA-seq, which may fail to capture transcripts due to limitations in depth of coverage. Moreover, immunohistochemistry can have limitations in detecting very low expression levels. Costa et al. 2013 (Translational Lung Cancer Research) states that “RNA-Seq technique will not likely replace current RT-PCR methods, but will be complementary depending on the needs and the resources as the results of the RNA-Seq will identify those genes that need to then be examined using RT-PCR methods”.

Similarly, regarding *Ir51b*, while the *GAL4* reporter lines from the Benton and Carlson labs do not show robust adult expression, our RT-PCR and functional data strongly support a role for *Ir51b* in labellar bitter GRNs. Specifically, *Ir51b^1^* mutants display electrophysiological deficits in response to cholesterol (Figure 2A–B), and these defects are rescued by expressing *Ir51b* in *Gr33a*-positive bitter neurons (Figure 3G), providing functional validation of the RT-PCR expression.

(4) The premise that high cholesterol intake is harmful to flies, which makes sensory mechanisms for cholesterol avoidance necessary, is interesting but underdeveloped. Animal sensory systems typically evolve to detect ecologically relevant stimuli with dynamic ranges matching environmental conditions. Given that *Drosophila* primarily consume fruits and plant matter (which contain minimal cholesterol) rather than animal-derived foods (which contain higher cholesterol), the ecological relevance of cholesterol detection requires more thorough discussion. Furthermore, at high concentrations, chemicals often activate multiple receptors beyond those specifically evolved for their detection. If the cholesterol concentrations used in this study substantially exceed those encountered in the fly's natural diet, the observed responses may represent an epiphenomenon rather than an ecologically and ethologically relevant sensory mechanism. What is the cholesterol content in flies' diet and how does that compare to the concentrations used in this paper?

*Drosophila melanogaster* cannot synthesize sterols *de novo,* and must acquire them from its diet. In natural environments, flies acquire sterols from fermenting fruit, decaying plant matter, and yeast, which contain trace amounts of phytosterols (e.g., stigmasterol, β-sitosterol) and ergosterol. While the exact sterol concentrations in these sources remain uncharacterized, our behavioral assays used concentrations (0.001–0.01% by weight) that align with the low levels expected in such nutrient-limited ecological niches.

In our study, the cholesterol concentrations tested ranged from 0.001% to 0.1%, thereby spanning both the physiologically relevant and slightly elevated range. Importantly, avoidance behaviors and receptor activation were most prominent at 0.1% cholesterol. While it is true that high chemical concentrations may elicit off-target effects via broad receptor activation, our genetic and electrophysiological data indicate that the observed responses are mediated by specific ionotropic receptors (*Ir51b*, *Ir7g*, *Ir56d*) and not merely generalized chemical stress.

Ecologically, elevated sterol levels may also signal conditions unsuitable for egg-laying or larval development. For example, high levels of cholesterol or other sterols may occur in substrates colonized by pathogenic microbes, decaying animal tissue, or in cases of abnormal microbial fermentation, which could represent a nutritional or microbial hazard. The avoidance of cholesterol may help signal the flies to avoid consuming decaying animal tissue. In this context, sensory detection of excessive cholesterol might serve as a protective function.

**Reviewer #2 (Public review):**
Summary:In Cholesterol Taste Avoidance in *Drosophila melanogaster*, Pradhan et al. used behavioral and electrophysiological assays to demonstrate that flies can: (1) detect cholesterol through a subset of bitter-sensing gustatory receptor neurons (GRNs) and (2) avoid consuming food with high cholesterol levels. Mechanistically, they identified five members of the IR family as necessary for cholesterol detection in GRNs and for the corresponding avoidance behavior. Ectopic expression experiments further suggested that Ir7g + Ir56d or Ir51b + Ir56d may function as tuning receptors for cholesterol detection, together with the Ir25a and Ir76b co-receptors.Strengths:The experimental design of this study was logical and straightforward. Leveraging their expertise in the *Drosophila* taste system, the research team identified the molecular and cellular basis of a previously unrecognized taste category, expanding our understanding of gustation. A key strength of the study was its combination of electrophysiological recordings with behavioral genetic experiments.Weaknesses:My primary concern with this study is the lack of a systematic survey of the IRs of interest in the labellum GRNs. Consequently, there is no direct evidence linking the expression of putative cholesterol IRs to the B GRNs in the S6 and S7 sensilla.Specifically, the authors need to demonstrate that the IR expression pattern explains cholesterol sensitivity in the B GRNs of S6 and S7 sensilla, but not in other sensilla. Instead of providing direct IR expression data for all candidate IRs (as shown for Ir56d in Figure 2-figure supplement 1F), the authors rely on citations from several studies (Lee, Poudel et al. 2018; Dhakal, Sang et al. 2021; Pradhan, Shrestha et al. 2024) to support their claim that Ir7g, Ir25a, Ir51b, and Ir76b are expressed in B GRNs (Lines 192-194). However, none of these studies provide GAL4 expression or in situ hybridization data to substantiate this claim.Without a comprehensive IR expression profile for GRNs across all taste sensilla, it is difficult to interpret the ectopic expression results observed in the B GRN of the I9 sensillum or the A GRN of the L-sensillum (Figure 4). It remains equally plausible that other tuning IRs-beyond the co-receptor Ir25a and Ir76b-could interact with the ectopically expressed IRs to confer cholesterol sensitivity, rather than the proposed Ir7g + Ir56d or Ir51b + Ir56d combinations.

We provide electrophysiological data demonstrating that the S6 and S7 sensilla respond to cholesterol (Figure 1D). This finding is consistent with the hypothesis that these sensilla harbor the complete receptor complexes necessary for cholesterol detection. In our electrophysiological recordings, only those bitter GRNs that co-express *Ir56d* along with either *Ir7g* or *Ir51b* generate action potentials in response to cholesterol. Other S-type sensilla lacking one or more of these subunits remain unresponsive, reinforcing the idea that these components are necessary for receptor function and sensory coding of cholesterol. Moreover, in the cholesterol-insensitive I9 sensillum (based on our mapping results using electrophysiology), co-expression of either *Ir7g + Ir56d* or *Ir51b + Ir56d* conferred *de novo* cholesterol sensitivity (Figure 4B). Importantly, no cholesterol response was observed when any of these *Ir*s was expressed alone or when *Ir7g + Ir51b* were co-expressed without *Ir56d*. These findings strongly argue against the possibility that endogenous tuning IRs in I9 sensilla (e.g., *Ir25a*, *Ir76b*) are sufficient to generate cholesterol responsiveness.

Furthermore, based on the literature, *Ir25a* and *Ir76b* are endogenously expressed in I- and L-type sensilla. Thus, their presence alone is insufficient for cholesterol responsiveness. These data support the model that cholesterol sensitivity depends on a specific, multi-subunit receptor complex (e.g., *Ir7g + Ir25a + Ir56d + Ir76b* or *Ir51b + Ir25a + Ir56d + Ir76b*).

In conclusion, while we acknowledge that our data do not provide a full anatomical map of *Ir* expression across all sensilla, our results strongly support the idea that cholesterol sensitivity in S6 and S7 sensilla arises from specific combinations of IRs expressed in the B GRNs.

**Reviewer #3 (Public review):**
Summary:Whether and how animals can taste cholesterol is not well understood. The study provides evidence that (1) cholesterol activates a subset of bitter-sensing gustatory receptor neurons (GRNs) in the fly labellum, but not other types of GRNs, (2) flies show aversion to high concentrations of cholesterol, and this is mediated by bitter GRNs, and (3) cholesterol avoidance depends on a specific set of ionotropic receptor (IR) subunits acting in bitter GRNs. The claims of the study are supported by electrophysiological recordings, genetic manipulations, and behavioral readouts.Strengths:Cholesterol taste has not been well studied, and the paper provides new insight into this question. The authors took a comprehensive and rigorous approach in several different parts of the paper, including screening the responses of all 31 labellar sensilla, screening a large panel of receptor mutants, and performing misexpression experiments with nearly every combination of the 5 IRs identified. The effects of the genetic manipulations are very clear and the results of electrophysiological and behavioral studies match nicely, for the most part. The appropriate controls are performed for all genetic manipulations.Weaknesses:The weaknesses of the study, described below, are relatively minor and do not detract from the main conclusions of the paper.(1) The paper does not state what concentrations of cholesterol are present in *Drosophila*'s natural food sources. Are the authors testing concentrations that are ethologically *Drosophila melanogaster* primarily feeds on fermenting fruits and associated microbial communities, especially yeast, which serve as major sources of dietary sterols. These natural food sources are known to contain phytosterols such as stigmasterol and β-sitosterol. One study quantified phytosterols (e.g., stigmasterol, sitosterol) in fruits, reporting concentrations between 1.6–32.6 mg/100 g edible portion (~0.0016–0.0326% wet weight) (Han et al 2008). The range we tested falls within this range. Additionally, ergosterol, the principal sterol in yeast and a structural analog of cholesterol, is present at levels of about 0.005% to 0.02% in yeast-rich environments.

To ensure physiological relevance, we designed our behavioral assays to include a broad concentration range of cholesterol, from 10^-5^% to 10^-1^%. This spans both physiological levels (0.001–0.01%), which are comparable to those found in the natural diet, and supra-physiological levels (e.g., 0.1%), which exceed natural exposure but help define the threshold for aversive behavior.

Our results demonstrate that flies begin to avoid cholesterol at concentrations ≥10^-3^% more (Figure 3A), which falls within the upper physiological range and may reflect the threshold beyond which cholesterol or related sterols become deleterious. At these higher concentrations, excess sterols may disrupt membrane fluidity, interfere with hormone signaling, or promote microbial overgrowth—all of which could compromise fly health.

(2) The paper does not state or show whether the expression of IR7g, IR51b, and IR56d is confined to bitter GRNs. Bitter-specific expression of at least some of these receptors would be necessary to explain why bitter GRNs but not sugar GRNs (or other GRN types) normally show cholesterol responses.

We show the *Ir56d-Gal4* is co-expressed with *Gr66a-GFP* in S6/S7 sensilla, indicating that it is expressed in bitter GRNs (Figure 2—figure supplement 1F). In the case of *Ir7g* and *Ir51b*, there are no reporters or antibodies to address expression. However, previously they have been shown to be expressed in bitter (B) GRNs using RT-PCR (Dhakal et al. 2021, Communications Biology; Pradhan et al. 2024, Journal of Hazardous Materials). In addition, we provide functional evidence that B GRNs are required for the cholesterol response since silencing B GRNs abolishes cholesterol-induced action potentials (Figure 1E–F). Moreover, we showed that we could rescue the *Ir7g^1^*, *Ir51b^1^* and *Ir56d^1^* mutant phenotypes only when we expressed the cognate transgenes in B GRNs using the *Gr33a-GAL4* (Figure 3G). Thus, while *Ir7g*/*Ir51b* are not exclusive to B GRNs, their functional role in cholesterol detection is B-GRN-specific.

(3) The authors only investigated the responses of GRNs in the labellum, but GRN responses in the leg may also contribute to the avoidance of cholesterol feeding. Alternatively, leg GRNs might contribute to cholesterol attraction that is unmasked when bitter GRNs are silenced. In support of this possibility, Ahn et al. (2017) showed that Ir56d functions in sugar GRNs of the leg to promote appetitive responses to fatty acids.

This is an interesting idea. Indeed, when bitter GRNs are hyperpolarized, the flies exhibit a strong attraction to cholesterol. Nevertheless, the cellular basis for cholesterol attraction and whether it is mediated by GRNs in the legs will require a future investigation.

(4) The authors might consider using proboscis extension as an additional readout of taste attraction or aversion, which would help them more directly link the labellar GRN responses to a behavioral readout. Using food ingestion as a readout can conflate the contribution of taste with post-ingestive effects, and the regulation of food ingestion also may involve contributions from GRNs on multiple organs, whereas organ-specific contributions can be dissociated using proboscis extension. For example, does presenting cholesterol on the proboscis lead to aversive responses in the proboscis extension assay (e.g., suppression of responses to sugar)? Does this aversion switch to attraction when bitter GRNs are silenced, as with the feeding assay?

We thank the reviewer for the suggestion regarding the use of the proboscis extension reflex (PER) assay to strengthen the link between labellar GRN activity and behavioral responses to cholesterol.

**Author response image 1. sa4fig1:** 

Our PER assay results shown above indicate that cholesterol presentation on the labellum or forelegs leads to an aversive response, as evidenced by a significant reduction in proboscis extension when compared to control stimuli (Author response image 1A. 2% sucrose or 2% sucrose with 10^-1^% cholesterol was applied to labellum or forelegs and the percent PER was recorded. n=6. Data were compared using single-factor ANOVA coupled with Scheffe’s post-hoc test. Statistical significance was compared with the control. Means ± SEMs. **p<0.01). This finding supports the idea that cholesterol is detected by labellar and leg GRNs and elicits behavioral avoidance. In contrast, sucrose stimulation robustly induces proboscis extension, as expected for an appetitive stimulus. We confirmed the defects of due to each *Ir* mutant by presenting the stimuli to the labellum (Author response image 1B). Together, these PER results provide a more direct behavioral correlate of labellar and leg GRN activation and reinforce our conclusion that cholesterol is sensed as an aversive tastant through the labellar bitter GRNs.

(5) The authors claim that the cholesterol receptor is composed of IR25a, IR76b, IR56d, and either IR7g or IR51b. While the authors have shown that IR25a and IR76b are each required for cholesterol sensing, they did not show that both are required components of the same receptor complex. If the authors are relying on previous studies to make this assumption, they should state this more clearly. Otherwise, I think further misexpression experiments may be needed where only IR25a or IR76b, but not both, are expressed in GRNs.

In our study, we relied on prior work demonstrating that *Ir25a* and *Ir76b* function as broadly required co-receptors in most IR-dependent chemosensory pathways (Ganguly et al., 2017; Lee et al., 2018). These studies showed that *Ir25a* and *Ir76b* are co-expressed in many GRNs across multiple taste modalities. Functional IR complexes often fail to form or signal properly in the absence of these co-receptors. Thus, it is widely accepted in the field that *Ir25a* and *Ir76b* function together as a core heteromeric scaffold for diverse IR complexes, akin to co-receptors in other ionotropic glutamate receptor families. We state that while *Ir25a* and *Ir76b* are presumed co-receptors in the cholesterol receptor complex based on their conserved roles, their direct physical interaction with *Ir7g*, *Ir51b*, and *Ir56d* remains to be demonstrated.

In support of this model, we note that in our ectopic expression experiments using I9 sensilla, which endogenously express *Ir25a* and *Ir76b*, introduction of either *Ir7g + Ir56d* or *Ir51b + Ir56d* was sufficient to confer cholesterol sensitivity (Figure 4B). We obtained a similar result in L6 sensilla (Figure 4D), which also endogenously express *Ir25a* and *Ir76b*. These findings imply that both co-receptors are already present in these sensilla and are likely part of the functional complex. However, we agree that we have not directly tested the requirement for both co-receptors in a minimal reconstitution context, such as expressing only *Ir25a* or *Ir76b* alongside tuning IRs in an otherwise null background. Such an experiment would indeed provide more direct evidence of their joint requirement in the receptor complex. Future studies, including heterologous expression experiments, will be necessary to define the cholesterol-receptor complexes.